# Breaking the Compression Ceiling: Data-Free Pipeline for Ultra-Efficient Delta Compression

**Xiaohui Wang**[1*]   **Peng Ye**[2,3*]   **Chenyu Huang**[1]   **Shenghe Zheng**[2]
**Bo Zhang**[2]   **Lei Bai**[2]   **Wanli Ouyang**[2,3]   **Tao Chen**[1,4†]
[1]Fudan University   [2]Shanghai AI Laboratory
[3]The Chinese University of Hong Kong   [4]Shanghai Innovation Institute
heatherwang000@gmail.com eetchen@fudan.edu.cn

## Abstract

With the rise of the fine-tuned–pretrained paradigm, storing numerous fine-tuned models for multi-tasking creates significant storage overhead. Delta compression alleviates this by storing only the pretrained model and the highly compressed delta weights (the differences between fine-tuned and pretrained model weights). However, existing methods fail to maintain both high compression and performance, and often rely on data. To address these challenges, we propose UltraDelta, the first data-free delta compression pipeline that achieves both ultra-high compression and strong performance. UltraDelta is designed to minimize redundancy, maximize information, and stabilize performance across inter-layer, intra-layer, and global dimensions, using three key components: (1) Variance-Based Mixed Sparsity Allocation assigns sparsity based on variance, giving lower sparsity to high-variance layers to preserve inter-layer information. (2) Distribution-Aware Compression applies uniform quantization and then groups parameters by value, followed by group-wise pruning, to better preserve intra-layer distribution. (3) Trace-Norm-Guided Rescaling uses the trace norm of delta weights to estimate a global rescaling factor, improving model stability under higher compression. Extensive experiments across (a) large language models (fine-tuned on LLaMA-2 7B and 13B) with up to $50\times$ compression, (b) general NLP models (RoBERTa-base, T5-base) with up to $224\times$ compression, (c) vision models (ViT-B/32, ViT-L/14) with up to $132\times$ compression, and (d) multi-modal models (BEiT-3) with $18\times$ compression, demonstrate that UltraDelta consistently outperforms existing methods, especially under ultra-high compression. Code is available at https://github.com/xiaohuiwang000/UltraDelta.

## 1   Introduction

As fine-tuning pretrained models for downstream tasks becomes increasingly popular, a growing number of task-specific fine-tuned models have been developed across various domains. To obtain multi-task capabilities and achieve optimal performance across tasks, deploying multiple fine-tuned models simultaneously has become a common practice. However, multi-model deployment introduces severe storage and computational overhead, since each fine-tuned model requires storing a full set of parameters. Delta Compression has emerged as a promising solution to this problem by substantially reducing storage requirements. Instead of storing multiple complete fine-tuned models, delta compression works by storing a single pretrained model along with a set of delta weights (i.e., the differences between each fine-tuned model and the pretrained model) and then applying aggressive compression to these delta weights to minimize storage overhead. These delta weights

---

[*]Equal Contribution.
[†]Corresponding Author.

39th Conference on Neural Information Processing Systems (NeurIPS 2025).

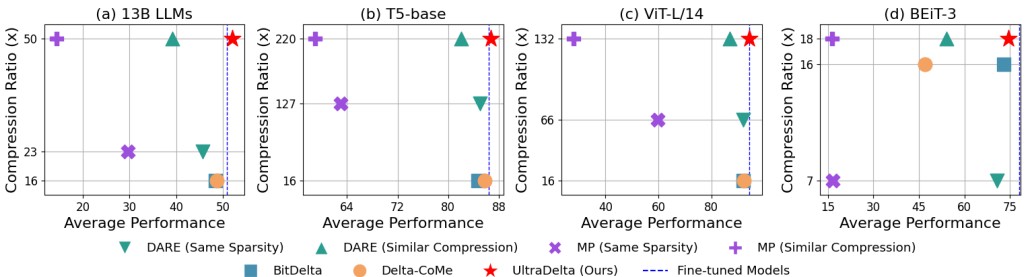

Figure 1: Compression vs. Performance. We compare UltraDelta with baselines, showing strong performance at ultra-high compression. DARE and Magnitude Pruning are evaluated in two settings: with the same sparsity as UltraDelta, and using pruning alone to match a similar compression ratio as UltraDelta. Subfigures show average performance on: (a) LLaMA-2-13B fine-tuned on 3 tasks, (b) T5-base on 8 NLP tasks, (c) ViT-L/14 on 8 vision tasks, and (d) BEiT-3 on 3 vision-language tasks.

are often highly redundant, allowing for substantial compression with minimal impact on model performance, thereby significantly reducing storage costs.

Currently, a number of works have been proposed for delta compression, which can be broadly categorized into two main types: Pruning and Quantization. Pruning methods reduce model size by removing parameters. For example, DARE [91] randomly prunes delta parameters, while Magnitude Pruning [27, 28, 46, 45] removes low-magnitude parameters. Quantization methods work by mapping full-precision parameters to low-bit representations. For example, BitDelta [53] uses masking and rescaling to achieve 1-bit quantization, while Delta-CoMe [64] applies mixed-precision quantization to matrices obtained from singular value decomposition. Other approaches, such as DeltaZip [90], combine quantization with pruning to improve deployment efficiency. However, these methods all struggle to balance ultra-high compression with strong performance, mainly due to: (1) Ignoring inter-layer differences: All layers are treated equally, ignoring that different layers contribute unequally to the model, which limits information preservation. (2) Disrupting intra-layer distributions: Intra-layer weight distributions are crucial to performance but are often distorted by aggressive quantization or pruning. (3) Lack of stability under ultra-high compression: Existing methods struggle to maintain stability under ultra-high compression without relying on data, leading to severe performance drops.

To address these limitations, we propose UltraDelta, the first data-free pipeline to achieve ultra-efficient delta compression, delivering both ultra-high compression and strong performance. UltraDelta focuses on minimizing parameter redundancy, maximizing information preservation, and enhancing model stability across three dimensions: inter-layer, intra-layer and global. It consists of three key components: (1) Variance-Based Mixed Sparsity Allocation (MSA, inter-layer): We theoretically show that layer-wise variance reflects the amount of information in each layer. Based on this insight, we assign lower sparsity to layers with higher variance to better preserve critical information. (2) Distribution-Aware Compression (DAC, intra-layer): We first apply uniform quantization, and group parameters by their quantized value, then perform random pruning within each group to maintain the relative proportions across different values and better preserve the original distribution. (3) Trace-Norm-Guided Rescaling (TNGR, global): We observe that under extreme sparsity, the standard rescaling factor $1/(1-s)$ becomes insufficient to stabilize performance (where $s$ is the sparsity rate). We introduce a refined rescaling factor $\gamma/(1-s)$, where $\gamma$ is heuristically estimated from the trace norm of each delta weight and is smaller for delta weights with larger trace norms, enhancing robustness under ultra-high compression.

We conduct extensive experiments across models of different scales, types, and tasks to evaluate the effectiveness and robustness of UltraDelta. The results demonstrate its exceptional performance under ultra-high compression: (1) Large Language Models: UltraDelta compresses LLaMA-2 series models [78, 56, 57] to 32× compression for 7B and 50× for 13B, consistently outperforming all baselines and even surpassing the average performance of fine-tuned models. To further demonstrate generality, we also evaluate on more recent architectures, including LLaMA-3.1 [43], Qwen2.5 [89], and Qwen3 [77]. (2) General NLP Models: UltraDelta achieves 224× compression on T5-base [67] and 32× on RoBERTa-base [54] across 8 NLP tasks, even exceeding the average performance of fine-tuned models on T5-base. (3) Vision Models: UltraDelta achieves 50× compression on ViT-B/32 and 132× on ViT-L/14 [66] across 8 image classification tasks, achieving completely lossless performance on ViT-L/14 compared with fine-tuned models. (4) Multi-modal Models: On

BEiT-3 [82], we achieve $18\times$ compression on 3 vision-language tasks, outperforming all baselines across all tasks. As illustrated in Fig. 1, we present compression–performance trade-off for all four model types. UltraDelta consistently achieves the best trade-off, outperforming all baselines, and in some cases, even surpassing the performance of fine-tuned models under ultra-high compression.

The main contributions of this paper can be summarized as follows:

• To break the compression ceiling of delta weights, we analyze the limitations of existing methods in information preservation and model stability, and propose UltraDelta, the first data-free pipeline enabling ultra-efficient delta compression, achieving both ultra-high compression ratios and strong performance without relying on any data.

• To enhance information preservation and model stability, we introduce three novel techniques: Variance-Based Mixed Sparsity Allocation to prioritize critical layers, Distribution-Aware Compression to preserve the original weight distribution, and Trace-Norm-Guided Rescaling to stabilize model performance under extreme sparsity.

• Extensive experiments across large language models, general NLP models, vision models, and multi-modal models demonstrate that UltraDelta consistently outperforms all baselines and even surpasses fine-tuned models in several settings, showcasing its effectiveness and robustness in achieving ultra-efficient delta compression.

## 2 Related Work

Delta Compression is a technique that compresses the delta weights. It is especially beneficial in multi-model deployment scenarios, where many task-specific models are fine-tuned from the same pretrained model. By storing only the highly compressed delta weights alongside the shared pretrained model, delta compression significantly eliminates redundant storage. Existing methods for delta compression primarily fall into two categories: pruning and quantization.

**Delta Weight Pruning** reduces model size by removing parameters from the delta weights. Magnitude Pruning (MP)[27, 28, 45, 46] removes weights with the smallest magnitudes but fails to preserve information and causes significant performance drops under high sparsity, as it disrupts the original weight distribution. DARE[91] applies random pruning combined with global rescaling based on sparsity to improve robustness, but suffers from instability under high sparsity. DAREx [20] refines DARE's rescaling using activation statistics, and DeltaDQ [38] performs group-wise pruning and selects the optimal group size based on attention error. However, both methods require data for tuning, limiting their use in data-free scenarios. Some methods also adopt layer-wise pruning [47, 55], but the metrics are often computationally expensive and usually require data. Overall, pruning effectively reduces parameter redundancy but suffers from several drawbacks, such as insufficient information preservation, instability under high sparsity, and reliance on additional data.

**Delta Weight Quantization** compresses delta weights by mapping them to low-bit representations. GPT-Zip[35] extends GPTQ[22] and quantizes delta parameters to 2-bit. Delta-DCT [33] leverages the discrete cosine transform and compresses delta weights in the frequency domain. BitDelta [53] uses binary masks and scaling factors for 1-bit quantization, but requires calibration data and also distorts the original weight distribution due to the binary masks. Delta-CoMe [64] applies singular value decomposition and mixed-precision quantization on the decomposed matrices to achieve 1-bit compression. Overall, although quantization effectively reduces parameter bit-width, it is often limited to 1-bit precision, which prevents further minimization of parameter redundancy.

**Hybrid Approach** combines delta weight pruning and delta weight quantization to leverage the advantages of both. DeltaZip [90] employs structured pruning and quantization, primarily aiming to improve deployment efficiency and hardware acceleration, but it falls short in preserving critical information, limiting its effectiveness under ultra-high compression. ComPEFT [88], on the other hand, applies magnitude pruning and then quantizes the remaining weights with a fixed shared value. However, it requires validation data and, due to the reliance on magnitude pruning, disrupts structural integrity and fails to preserve critical information under high sparsity.

In contrast, UltraDelta is a data-free hybrid compression method that explicitly focuses on minimizing parameter redundancy, maximizing information preservation, and enhancing model stability under ultra-high compression ratios, all without requiring any data.

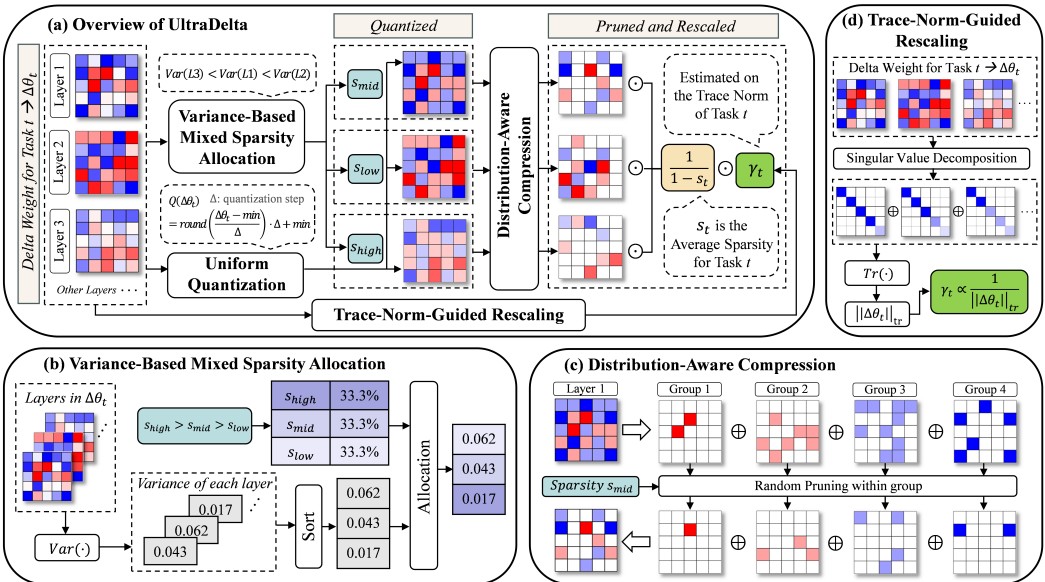

Figure 2: (a) Overview of the UltraDelta pipeline. It first assigns layer-wise sparsity based on variance, and performs uniform quantization followed by group-wise pruning, and finally applies trace-norm-guided rescaling. (b) Variance-Based Mixed Sparsity Allocation allocates lower sparsity to high-variance layers to preserve information. (c) Distribution-Aware Compression groups quantized parameters by value and prunes within each group to retain distribution. (d) Trace-Norm-Guided Rescaling estimates a rescaling factor using the trace norm of each delta weight to enhance robustness.

# 3 Method

## 3.1 Preliminaries

Consider a set of $T$ fine-tuned models that are fine-tuned from the same pretrained model. Let the weights of these models be denoted as $\{\theta_1, \theta_2, \ldots, \theta_T\}$, and the shared pretrained model weight as $\theta_{pre}$. For each fine-tuned model $t \in \{1, 2, \ldots, T\}$, we define the delta weight as:

$$\Delta\theta_t = \theta_t - \theta_{\mathrm{pre}} \tag{1}$$

## 3.2 UltraDelta: A Data-Free Pipeline for Ultra-Efficient Delta Compression

To address the limitations of existing delta compression methods, UltraDelta introduces a hybrid compression framework that minimizes parameter redundancy while maximizing information preservation and enhancing model stability across three dimensions: inter-layer, intra-layer, and global, enabling robustness under ultra-high compression. An overview of the overall pipeline is illustrated in Fig. 2. We detail the core design of UltraDelta in Sec.3.2, and provide analysis in Sec.3.3.

**(1) Variance-Based Mixed Sparsity Allocation (MSA).** Existing pruning methods often adopt uniform sparsity across layers, treating all parts of the model equally and overlooking their varying importance. However, different layers exhibit different sensitivities to pruning. We observe that layers with higher variance tend to carry more critical information and are more sensitive to pruning (see Sec. 3.3). Motivated by this, we propose MSA that categorizes all layers into three groups based on their variances (low, medium, high), each containing an equal number of parameters:

$$\mathrm{Groups} = \{\mathcal{G}_{\mathrm{low}}, \mathcal{G}_{\mathrm{mid}}, \mathcal{G}_{\mathrm{high}}\} \tag{2}$$

Each group $\mathcal{G}_v$ is assigned a sparsity rate $s_v$ based on its variance $v \in \{\mathrm{low}, \mathrm{mid}, \mathrm{high}\}$, with lower-variance groups assigned higher sparsity and higher-variance groups assigned lower sparsity, in order to preserve inter-layer information better:

$$s_v = s_{\mathrm{mid}} + \delta_v, \quad \delta_v \in \{+s_{\mathrm{step}}, 0, -s_{\mathrm{step}}\} \tag{3}$$

where $s_{\mathrm{mid}}$ is the average sparsity of the overall delta weight, and $s_{\mathrm{step}} > 0$ controls the difference in sparsity across groups.

**(2) Distribution-Aware Compression (DAC).** Prior work has shown that preserving the shape of the weight distribution after compression helps maintain model performance [33]. To better preserve the intra-layer distribution shape under ultra-high compression, we propose DAC. Specifically, as delta weight exhibits a distribution well-suited to uniform quantization [38], we first apply uniform quantization with a lower bit width $b$ (typically $b = 4$) to the delta weight. This quantization process consists of two steps: (a) mapping each value in $\Delta\theta_t$ to a discrete integer within $[0, 2^b - 1]$; (b) mapping the quantized values back to the original value range. Let $\min = \min(\Delta\theta_t)$, $\max = \max(\Delta\theta_t)$, and $\Delta$ be the quantization step size:

$$\hat{\Delta\theta}_t = \text{round}\left(\frac{\Delta\theta_t - \min}{\Delta}\right) \cdot \Delta + \min, \quad \text{where} \quad \Delta = \frac{\max - \min}{2^b - 1} \tag{4}$$

After quantization, we group the parameters by their values and perform random pruning within each group. Let $\{u_1, u_2, \ldots, u_n\}$ represent the unique values in the quantized delta weight $\hat{\Delta\theta}_t$. We denote $(j, k)$ as the row and column indices of elements in $\hat{\Delta\theta}_t$. For each unique quantized value $u_i$, we define a mask $\boldsymbol{M}_{u_i}$ whose elements are independently sampled from a Bernoulli distribution with success probability $1 - s_l$, where $s_l$ is the sparsity rate of the layer. Specifically:

$$\boldsymbol{M}_{u_i}^{(j,k)} \sim \begin{cases} \text{Bernoulli}(1 - s_l), & \text{if } \hat{\Delta\theta}_t^{(j,k)} = u_i \\ 0, & \text{otherwise} \end{cases} \tag{5}$$

The pruned delta weight $\hat{\Delta\theta}_t^*$ is obtained by applying the group-wise Bernoulli masks to the quantized delta weights:

$$\hat{\Delta\theta}_t^* = \sum_{i=1}^n u_i \cdot \boldsymbol{M}_{u_i} \tag{6}$$

DAC preserves the relative proportions of delta values, maintaining the intra-layer distribution shape. It minimizes distortion under high sparsity and ensures more stable performance. Moreover, DAC can be applied in non-quantized settings with minimal modification, making it suitable for scenarios where aggressive compression is not required. Instead of grouping by discrete values, we partition the weight range into several intervals, and assign parameters whose values fall within the same interval to the same group. Details are provided in App. A.

**(3) Trace-Norm-Guided Rescaling (TNGR).** As derived in Sec. 3.3, the variance of activation errors between compressed and original delta weights correlates with $s/(1 - s)$, causing instability at high sparsity. To address this, we introduce an additional rescaling factor $\gamma_t$ for task $t$, and redefine rescaling as $\gamma_t/(1 - s_t)$, where $s_t$ is the overall sparsity of the delta weight for task $t$. We find that delta weights with larger trace norms (i.e., the sum of their singular values) tend to require smaller $\gamma_t$ to maintain stable performance. Motivated by this observation, we set $\gamma_t$ inversely proportional to the trace norm of the delta weights:

$$\gamma_t \propto \frac{1}{||\Delta\theta_t||_{\text{tr}}} \tag{7}$$

where $||\cdot||_{\text{tr}}$ denotes the trace norm. This trace-norm-guided rescaling offers a simple yet effective way to adaptively stabilize pruning under extreme sparsity. In practice, $\gamma_t$ typically falls within the range of $[0.5, 1.0]$. During the inference stage, the final model weight is reconstructed as:

$$\theta_t^{\text{final}} = \theta_{\text{pre}} + \frac{\gamma_t}{1 - s_t} \cdot \hat{\Delta\theta}_t^* \tag{8}$$

### 3.3 Reason for Effectiveness

#### 3.3.1 Theoretical Analysis of MSA

**In lossless compression,** the entropy represents the lower bound on the achievable average bit rate. A larger entropy means that a layer carries more information and therefore should be more carefully preserved. We show that the variance of a layer is closely related to its entropy, serving as the theoretical motivation for MSA. Following [50], we model the distribution of a layer as a Gaussian Distribution $L \sim \mathcal{N}(\mu, \sigma^2)$. The probability density function of $L$ is given by:

$$p(l) = \frac{1}{\sqrt{2\pi\sigma^2}} \exp\left(-\frac{(l - \mu)^2}{2\sigma^2}\right) \tag{9}$$

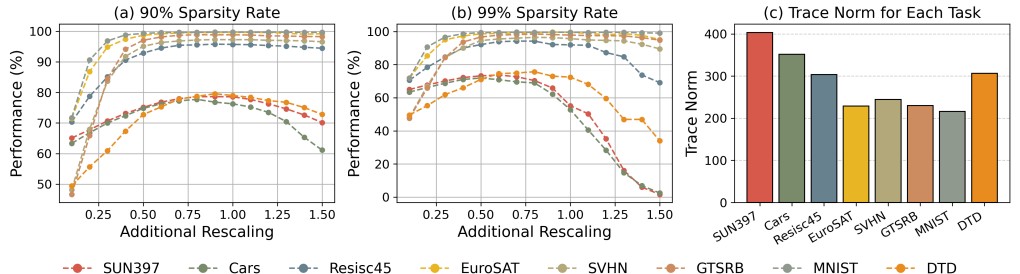

Figure 3: Relationship between Trace Norm and Additional Rescaling Factor on 8 ViT-B/32. (a) Average performance under 90% sparsity with different additional rescaling factors. (b) Same as (a), under 99% sparsity. (c) Trace norm of delta weights across tasks.

The entropy of this distribution is (detailed derivation in App. B.1):

$$H(L) = -\mathbb{E}[\log p(l)] = \log(\sigma) + \frac{1}{2}\log(2\pi) + \frac{1}{2} \tag{10}$$

This demonstrates the link between variance and entropy: layers with larger variance have higher entropy, implying greater information content and thus requiring smaller sparsity to preserve it.

**In lossy compression,** entropy alone is not sufficient. When distortion is allowed, we employ rate–distortion theory [6, 17], which establishes the limit on how much information must be retained to achieve a given distortion level. Let $\hat{L}$ denote the compressed version of $L$. With mean squared error (MSE) distortion, we define:

$$D = \mathbb{E}\big[(L - \hat{L})^2\big], \tag{11}$$

According to rate–distortion theory, the rate–distortion function of a Gaussian source under MSE distortion (which is independent of the mean $\mu$) is given by:

$$R(D) = \begin{cases} \frac{1}{2}\log\left(\frac{\sigma^2}{D}\right), & 0 < D < \sigma^2, \\ 0, & D \geq \sigma^2, \end{cases} \tag{12}$$

This shows that, for a fixed distortion $D$, a larger variance $\sigma^2$ leads to a higher rate $R(D)$, meaning that more information must be retained. In other words, layers with larger variance require a smaller sparsity in order to preserve their information under lossy compression.

### 3.3.2 Theoretical Analysis of the Rescaling Factor.

For a given delta weight $\Delta\theta$, let the activation be defined as $a = \Delta\theta \odot x$, where $x$ is the input feature. We introduce a Bernoulli mask $B \sim \text{Bernoulli}(1 - s)$, with sparsity rate $s$, and apply an additional scaling factor $\gamma \in [0, 1]$. Following [20], the activation error introduced by compression is:

$$\varepsilon = \Delta\theta \odot x - \frac{\gamma}{1 - s} \cdot (B \odot \Delta\theta \odot x) = a \odot \left(1 - B \cdot \frac{\gamma}{1 - s}\right) \tag{13}$$

We further derive the variance of this error (detailed derivation in App. B.2):

$$\text{Var}(\varepsilon) = \frac{\gamma^2 s}{1 - s} \odot a^2 \tag{14}$$

As sparsity becomes higher, the variance grows rapidly and causes instability. This underscores the importance of using smaller rescaling factors to stabilize the model under high sparsity.

### 3.3.3 Empirical Analysis of TNGR.

We investigate how the additional rescaling factor $\gamma$ for each delta weight correlates with its intrinsic characteristics, as shown in Fig. 3. At 90% sparsity, we observe that delta weights with larger trace norms tend to require smaller values of $\gamma$ to maintain stable performance. Moreover, their performance is more sensitive to changes in $\gamma$. This phenomenon becomes even more pronounced at 99% sparsity, where the need for smaller $\gamma$ is more substantial for delta weights with large trace norms, and the performance drop-off becomes steeper when $\gamma$ is not well matched. These empirical trends support our trace-norm-guided heuristic estimation, which sets $\gamma \in [0.5, 1.0]$ inversely proportional to the trace norm for adaptive and data-free rescaling.

### 3.3.4 Ability to Preserve Information.

To explain why UltraDelta achieves strong performance under ultra-high compression ratios, we analyze how well it preserves the original model's internal representations. Specifically, we compare the cosine similarity between the embeddings from the compressed and fine-tuned RoBERTa-base [54] models on the CoLA [83] dataset. We extract the layer-wise embeddings of each input token and report the average cosine similarities. As shown in Fig. 4, our method consistently maintains high similarity across

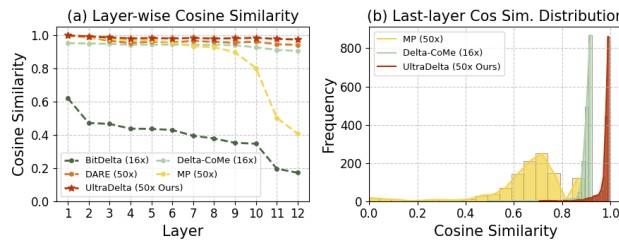

Figure 4: Cosine Similarity Analysis. (a) Layer-wise similarity of each layer's embeddings. (b) Distribution of cosine similarity for the final-layer embeddings. Compression ratios for each method are indicated in parentheses.

all layers, especially in the deeper ones that are more critical to performance. The distribution of cosine similarities in the final layer further confirms that UltraDelta retains essential information, indicating its effectiveness in preserving critical information despite ultra-high compression.

## 4 Experiment

**Baseline Methods.** We compare UltraDelta with the following baselines: (1) Fine-tuned Models; (2) Multi-task Learning (MTL) [8], which trains a single model on all tasks; (3) BitDelta [53]; (4) Delta-CoMe [64]; (5) DARE [91] with either the same sparsity or similar compression ratio as UltraDelta, using pruning alone to achieve a similar compression ratio; (6) Magnitude Pruning (MP) [27], also matched by sparsity or compression ratio. Since BitDelta and Delta-CoMe target only linear layers in Transformer blocks, we restrict all methods to compress only linear layers for fairness. All original models are stored in FP16 format. Experiments are conducted on a NVIDIA A800 GPU.

**Compression Ratio.** Unlike quantization, where compression is directly determined by bit-width, pruning methods require storing the indices of non-zero values. To ensure a unified and fair evaluation, following [70, 74, 88], we adopt Golomb coding [25] to store the compressed delta weights for methods involving pruning, which is well-suited for encoding the zero-run lengths that approximately follow a geometric distribution. Detailed derivation and calculations are in App. B.3, with results summarized in Tab. 8. Furthermore, for a complete evaluation, we present the ideal upper bound of compression ratio and practical results under different storage schemes in Sec. 4.5(storage scheme).

### 4.1 Performance on Large Language Models (LLMs)

**Settings.** We primarily evaluate UltraDelta on the LLaMA-2 series with sizes of 7B and 13B across three types of models: math (WizardMath [56]), code (WizardCoder [57]), and chat (LLAMA-2-Chat [78]). The models are evaluated using GSM8K [15] for math (accuracy), HumanEval [11] for code (pass@1), and TruthfulQA [49] for chat (accuracy). We also incorporate more recent LLMs and more challenging tasks: LLaMA-3.1-Tulu-8B [43] evaluated on MBPP+ [3] and HumanEval+ [11], Qwen2.5-7B-Instruct [89] evaluated on MATH [31] and GPQA [69], and Qwen3Guard-8B [77] evaluated on MMLU [30] and BBQ [63]. Details for the LLM IDs are presented in App. C.1.

**Results.** We first report results on the LLaMA-2 series (see Tab. 1). For the 7B model, we apply 4-bit quantization and prune 95% of parameters, achieving an 32.9× compression ratio; for the 13B model, we prune 97% to reach 50.9× compression. Despite these ultra-high compression ratios, UltraDelta achieves average scores of 45.57 (7B) and 52.05 (13B), exceeding the fine-tuned models (45.37 and 50.94, respectively). This improvement suggests that UltraDelta may introduce a regularization benefit. Compared to BitDelta [53] and Delta-CoMe [64], which are limited to 16× compression, UltraDelta delivers both higher compression and better performance. Other approaches such as DARE [91] and MP [27] fall short even at lower compression ratios, and when matched to the similar compression ratio as UltraDelta, their performance drops sharply. For newer LLMs (see Tab. 2), UltraDelta achieves the best overall compression–performance trade-off across baselines. In summary, UltraDelta consistently outperforms all baselines across various tasks and model sizes, demonstrating its remarkable robustness and effectiveness under ultra-high compression.

Table 1: Performance comparison on large language models across 3 tasks. Ratio denotes the compression ratio (original model size / compressed size). Methods marked with "(Sparsity)" use the same sparsity rate as ours, while those marked with "(Compression)" use a higher sparsity rate to match a similar overall compression ratio as our method. Best results are highlighted in bold.

| Method | Ratio | | WizardMath-V1.0 | | WizardCoder-V1.0 | | LLAMA-2-Chat | | Avg | |
|---|---|---|---|---|---|---|---|---|---|---|
| | 7B | 13B | 7B | 13B | 7B | 13B | 7B | 13B | 7B | 13B |
| Individual | 1× | 1× | 41.55 | 47.31 | 49.40 | 58.50 | 45.17 | 47 | 45.37 | 50.94 |
| BitDelta [53] | 16× | 16× | 36.54 | 46.12 | 46.30 | 54.30 | 42.84 | 44.85 | 41.89 | 48.42 |
| Delta-CoMe [64] | 16× | 16× | 37.25 | 46.87 | 45.30 | 53.70 | 45.02 | 45.57 | 42.52 | 48.71 |
| DARE [91] (Sparsity) | 14.7× | 23.7× | 36.62 | 46.70 | 46.20 | 53.90 | 33.54 | 36.29 | 38.79 | 45.63 |
| MP [27] (Sparsity) | 14.7× | 23.7× | 23.65 | 3.11 | 45.70 | 47.60 | 40.02 | 38.25 | 36.46 | 29.65 |
| DARE [91] (Compression) | 31.7× | 50.1× | 29.49 | 35.63 | 43.30 | 51.80 | 29.13 | 33.25 | 33.97 | 40.23 |
| MP [27] (Compression) | 31.7× | 50.1× | 16.22 | 8.42 | 32.90 | 16.50 | 31.95 | 21.15 | 27.02 | 15.36 |
| **UltraDelta (Ours)** | **32.9×** | **50.9×** | **38.76** | **48.45** | **51.80** | **59.10** | **46.14** | **48.59** | **45.57** | **52.05** |

Table 2: Performance comparison on recent large language models.

| Method | Ratio | Llama-3.1-Tulu-3-8B | | Qwen2.5-7B-Instruct | | Qwen3Guard-Gen-8B | | Avg |
|---|---|---|---|---|---|---|---|---|
| | | HumanEval+ | MBPP+ | MATH | GPQA | MMLU | BBQ | |
| Individual | 1× | 57.30 | 56.60 | 36.20 | 36.87 | 72.80 | 50.34 | 51.69 |
| BitDelta [53] | 16× | 55.50 | 52.90 | 38.10 | 37.37 | 73.05 | 49.60 | 51.09 |
| Delta-CoMe [64] | 16× | 53.80 | 53.10 | 29.45 | 31.43 | 71.46 | 44.61 | 47.31 |
| DARE [91] (Sparsity) | 14.7× | 50.60 | 50.00 | 37.76 | 34.85 | 72.18 | 49.69 | 49.18 |
| MP [27] (Sparsity) | 14.7× | 45.10 | 40.20 | 11.84 | 31.82 | 70.91 | **50.18** | 41.68 |
| **UltraDelta (Ours)** | **32.9×** | **56.10** | **54.50** | **40.14** | **38.38** | **73.44** | 49.74 | **52.05** |

## 4.2 Performance on general NLP models

**Settings.** Following [76, 91], we evaluate both T5-base [67] and RoBERTa-base [54] models on the GLUE [81] benchmark, covering CoLA [83], SST-2 [71], MRPC [21], STS-B [10], QQP [36], MNLI [84], QNLI [68], and RTE [24]. For T5-base, we report Spearman's $\rho$ on STS-B and accuracy on the other tasks. Settings and results of RoBERTa-base are provided in App. C.3.

**Results.** The results for T5-base models are presented in Tab. 3. Using 4-bit quantization combined with 99.5% pruning, UltraDelta achieves an impressive 224.6× compression ratio. Despite this extreme compression, UltraDelta attains an average accuracy of 86.74, outperforming all existing compression baselines and even slightly exceeding the full fine-tuned model's accuracy of 86.37. Compared to BitDelta [53] and Delta-CoMe [64], UltraDelta achieves a much higher compression ratio while still maintaining superior performance. This demonstrates that UltraDelta successfully pushes the compression limits of delta weights while preserving model performance, highlighting its effectiveness and strong generalization ability.

Table 3: Performance comparison on T5-base models across 8 tasks.

| Methods | Ratio | CoLA | SST2 | MRPC | STSB | QQP | MNLI | QNLI | RTE | Avg |
|---|---|---|---|---|---|---|---|---|---|---|
| Individual | 1× | 74.98 | 93.58 | 87.50 | 88.70 | 85.37 | 83.41 | 91.49 | 85.92 | 86.37 |
| BitDelta [53] | 16× | 70.09 | 93.46 | 84.06 | 86.20 | 85.26 | **83.64** | 90.94 | 83.75 | 84.68 |
| Delta-CoMe [64] | 16× | 74.26 | 93.58 | 87.01 | 87.94 | 85.44 | 83.51 | 91.54 | 84.36 | 85.95 |
| DARE [91] (Sparsity) | 127.6× | 74.16 | 93.34 | 87.83 | 87.86 | 85.22 | 83.19 | 91.52 | 84.11 | 85.90 |
| MP [27] (Sparsity) | 127.6× | 37.87 | 88.76 | 70.59 | 0 | 80.4 | 68.55 | 82.88 | 82.67 | 63.97 |
| DARE [91] (Compression) | 220.5× | 71.57 | 92.25 | 85.03 | 86.77 | 84.26 | 81.16 | 90.50 | 82.81 | 84.30 |
| MP [27] (Compression) | 220.5× | 30.87 | 87.84 | 70.83 | 0 | 79.16 | 56.36 | 76.31 | 82.17 | 60.44 |
| **UltraDelta (Ours)** | **224.6×** | **76.51** | **93.81** | **88.48** | **88.82** | **85.41** | 83.17 | **91.76** | **85.92** | **86.74** |

## 4.3 Performance on Vision Models

**Settings.** Following [34], we evaluate ViT-B/32 and ViT-L/14 [66] models on eight image classification datasets: SUN397 [87], Stanford Cars [40], RESISC45 [12], EuroSAT [29], SVHN [60], GTSRB [73], MNIST [44], and DTD [13]. Accuracy is used as the evaluation metric for all datasets.

**Results.** The results of ViT-L/14 models are shown in Tab. 4, while ViT-B/32 results are provided in App. C.2. With 4-bit quantization and 99% sparsity, UltraDelta achieves a 132.5× compression ratio while maintaining an average accuracy of 94.4, matching the full fine-tuned model and significantly outperforming all compressed baselines, demonstrating its effectiveness on vision models.

Table 4: Performance comparison on ViT-L/14 models across 8 tasks.

| Methods | Ratio | SUN397 | Cars | RESISC45 | EuroSAT | SVHN | GTSRB | MNIST | DTD | Avg |
|---------|-------|--------|------|----------|---------|------|-------|-------|-----|-----|
| Individual | 1× | 84.8 | 92.3 | 97.4 | 99.7 | 98.1 | 99.2 | 99.7 | 84.1 | 94.4 |
| Traditional MTL [8] | 8× | 80.8 | 90.6 | 96.3 | 96.3 | 97.6 | 99.1 | 99.6 | 84.4 | 93.1 |
| BitDelta [53] | 16× | 84.0 | 92.1 | 97.2 | 99.7 | 97.9 | 99.0 | 99.7 | 83.0 | 94.1 |
| Delta-CoMe [64] | 16× | **84.3** | 92.1 | 97.2 | 99.6 | 98.0 | 99.1 | 99.7 | 83.6 | 94.2 |
| DARE [91] (Sparsity) | 66.5× | 82.2 | 90.0 | 97.0 | 99.7 | 98.0 | 99.1 | 99.7 | 83.1 | 93.6 |
| MP [27] (Sparsity) | 66.5× | 26.7 | 43.9 | 83.4 | 92.8 | 24.4 | 62.9 | 77.7 | 66.1 | 59.7 |
| DARE [91] (Compression) | 127.6× | 65.2 | 78.4 | 94.2 | 99.6 | 97.7 | 99.2 | 99.7 | 79.1 | 89.1 |
| MP [27] (Compression) | 127.6× | 4.9 | 4.2 | 42.6 | 83.2 | 6.4 | 16.7 | 27.2 | 50.7 | 29.5 |
| **UltraDelta (Ours)** | **132.5×** | 84.1 | **92.2** | **97.4** | **99.8** | **98.1** | **99.2** | **99.8** | **84.3** | **94.4** |

## 4.4 Performance on multi-modal models

**Settings.** We compress delta weights on BEiT-3 [82] models fine-tuned on three datasets: VQA [26] (Visual Question Answering), NLVR2 [75] (Visual Reasoning), and COCO Captioning [51] (Image Captioning). The COCO Captioning task is evaluated using BLEU4 [61], CIDEr [79], METEOR [4], and ROUGE-L [48], while the other two tasks are evaluated based on accuracy.

**Results.** The results of BEiT-3 models are shown in Tab. 5. With 4-bit quantization and 90% sparsity, UltraDelta achieves a 18.4× compression ratio maintaining high accuracy and quality, outperforming all baselines across all tasks, demonstrating its effectiveness on multi-modal models.

Table 5: Performance comparison on multi-modal BEiT-3 models across 3 vision-language tasks.

| Methods | Task Metric | Ratio $\alpha(\uparrow)$ | COCO-Captioning | | | | NLVR2 Accuracy($\uparrow$) | VQAv2 Accuracy($\uparrow$) |
|---------|-------------|-------|---------|-------|--------|----------|-------|-------|
| | | | BLEU4($\uparrow$) | CIDEr($\uparrow$) | METEOR($\uparrow$) | ROUGE-L($\uparrow$) | | |
| Individual | 1× | | 0.394 | 1.337 | 0.331 | 0.601 | 84.269 | 83.49 |
| BitDelta [53] | 16× | | 0.367 | 1.250 | 0.294 | 0.581 | 82.317 | 74.59 |
| Delta-CoMe [64] | 16× | | 0.224 | 0.846 | 0.251 | 0.488 | 60.098 | 38.64 |
| DARE [91] (Sparsity) | 7.7× | | 0.333 | 1.15 | 0.285 | 0.564 | 80.652 | 73.29 |
| MP [27] (Sparsity) | 7.7× | | 0 | 0 | 0.006 | 0.014 | 49.263 | 0 |
| DARE [91] (Compression) | 18.1× | | 0.178 | 0.589 | 0.196 | 0.441 | 69.470 | 58.89 |
| MP [27] (Compression) | 18.1× | | 0 | 0 | 0.006 | 0.013 | 48.931 | 0.01 |
| **UltraDelta (Ours)** | **18.4×** | | **0.372** | **1.270** | **0.296** | **0.583** | **83.163** | **75.66** |

## 4.5 Ablation Study

**Effectiveness of Each Component.** We evaluate the contribution of each component on 8 ViT-B/32 in Tab. 6. DAC substantially improves compression ratio from 23.7× to 50.9×, all while preserving model accuracy. MSA yields the largest accuracy improvement, highlighting the effectiveness of mixed sparsity allocation. Together, they enable ultra-high compression with strong performance.

**Effectiveness of DAC.** To further validate the robustness of DAC, we evaluate it on a large-scale benchmark [32] of 30 ViT-B/16 models across 4-bit, 8-bit, and non-quantized configurations. For fair comparison, we apply uniform quantization to both methods in quantized cases. As shown in Fig. 5 (detailed results are in App. C.4.1), DAC consistently outperforms

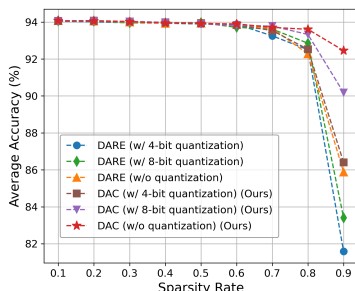

Figure 5: Average Performance of DAC and DARE on 30 ViT-B/16.

DARE [91] across all configurations. Notably, under 90% sparsity and 4-bit quantization, DAC still exceeds DARE's non-quantized performance, confirming its robustness across diverse tasks.

Table 6: Ablation study on the effectiveness of each component.

| Methods | SUN397 | Cars | RESISC45 | EuroSAT | SVHN | GTSRB | MNIST | DTD | Ratio | Avg |
|---|---|---|---|---|---|---|---|---|---|---|
| DARE [91] (97% Sparsity) | 76.0 | 73.3 | 95.1 | 99.7 | 97.0 | 98.5 | 99.6 | 78.1 | 23.7× | 89.7 |
| + DAC | 75.6 | 73.3 | 95.3 | 99.5 | 97.1 | 98.5 | 99.6 | 78.5 | 50.9× (↑ 27.2×) | 89.7 |
| + DAC + MSA | 76.9 | 75.9 | 95.5 | 99.2 | 97.0 | 98.8 | 99.7 | 79.2 | 50.9× | 90.3 (↑ 0.6) |
| + DAC + MSA + TNGR | 78.6 | 77.4 | 95.7 | 99.4 | 97.0 | 98.7 | 99.7 | 79.0 | 50.9× | 90.7 (↑ 0.4) |

Table 7: Compression ratios across storage formats and models.

| Format | LLaMA-2-7B | LLaMA-2-13B | ViT-B/32 | ViT-L/14 | T5-base | RoBERTa-base | BEiT-3 |
|---|---|---|---|---|---|---|---|
| Index-Free Storage | 80.0× | 133.0× | 133.0× | 400.0× | 800.0× | 80.0× | 40.0× |
| Golomb Coding | 32.9× | 50.9× | 50.9× | 132.5× | 224.6× | 32.9× | 18.4× |
| CSR | 19.5× | 30.7× | 35.4× | 105.2× | 188.3× | 21.4× | 10.8× |
| BCSR | 12.7× | 19.8× | 21.7× | 63.7× | 121.4× | 13.4× | 7.3× |

**Impact of Hyper-Parameters.** We evaluate key hyper-parameters in DAC and MSA on 8 ViT-B/32 models, as shown in Fig. 6 (detailed results are in App. C.4.2). For bit-width in quantization (see Eq. 4), accuracy improves notably from 2-bit to 4-bit, with marginal gains beyond, suggesting 4-bit as an optimal balance. For sparsity step $s_{\text{step}}$ in MSA (see Eq. 3),

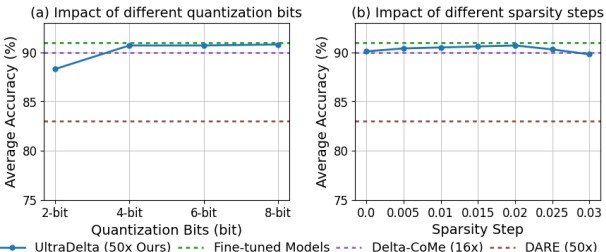

Figure 6: Impact of hyper-parameter on 8 ViT-B/32. (a) Impact of quantization bits; (b) Impact of sparsity steps.

we evaluate it under a fixed 97% target sparsity and 4-bit quantization. A moderate step size (within $[0.01, 0.02]$) yields the best average performance. Performance remains stable as long as extremely large or small step sizes are avoided. This indicates that precise tuning of $s_{\text{step}}$ is generally unnecessary.

**Storage Scheme.** We report compression ratios of UltraDelta under different storage schemes. Index-Free Storage denotes the ideal upper bound where only non-zero values are stored without indices (the ideal compression ratios of other pruning baselines are in Tab. 8). We also evaluate practical schemes, including Golomb coding, Compressed Sparse Row (CSR), and Block Compressed Sparse Row (BCSR). As shown in Tab. 7, BCSR performs poorly, as it only works when zeros are densely clustered and thus is not suitable for our setting.

## 5 Discussion and Conclusion

**Regularization Effect.** By controlling the fitting degree of models, we examine that UltraDelta particularly benefits underfitted models (see App. D.1 for detailed results and analysis). Since LLMs are typically evaluated in the zero-shot or few-shot settings, where underfitting is common, this explains why UltraDelta often yields not only strong compression but also performance gains.

**Overhead.** UltraDelta introduces only modest computational overhead during compression and is efficient at inference compared with baselines. The main cost arises from computing trace norms. Since only their relative magnitudes are required, fast approximation such as randomized SVD can be employed to reduce the overhead. In the inference stage, delta weights only need to be dequantized and added back to the base model, which can be done on CPU efficiently. In contrast, methods such as Delta-CoMe [64] require additional matrix multiplications, making them less efficient.

**Limitations and Future Work.** We acknowledge that our method employs a heuristic rescaling strategy and does not explicitly target deployment efficiency; detailed discussion is in App. D.3.

**Conclusion.** In this paper, we present UltraDelta, the first data-free pipeline for ultra-efficient delta compression, capable of achieving both ultra-high compression ratios and strong performance. UltraDelta is a hybrid compression method with a focus on preserving information across inter-layer, intra-layer, and global dimensions through three novel components. Extensive experiments on language, vision, and multi-modal models, along with theoretical analysis, validate the effectiveness, robustness, and generality of UltraDelta across diverse settings.

# 6    Acknowledgement

This work is supported by National Key Research and Development Program of China (No. 2022ZD0160101), Shanghai Natural Science Foundation (No. 23ZR1402900), Shanghai Science and Technology Commission Explorer Program Project (24TS1401300), Shanghai Municipal Science and Technology Major Project (No.2021SHZDZX0103). The computations in this research were performed using the CFFF platform of Fudan University.

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

# Appendix for UltraDelta

## A  Non-quantized Variant of Distribution-Aware Compression

While our main Distribution-Aware Compression operates on quantized delta weights, it can also be adapted to a non-quantized variant for scenarios where quantization is not applied or not desired. In this version, we replace the discrete grouping based on quantized values with continuous interval-based grouping over the full-precision delta weights.

Specifically, for a given delta weight $\Delta\theta_t$, we first compute its minimum and maximum values, denoted as $\min(\Delta\theta_t)$ and $\max(\Delta\theta_t)$. We then divide the range $[\min(\Delta\theta_t), \max(\Delta\theta_t)]$ into $I$ equally spaced intervals. The width of each interval is given by:

$$\delta_{\text{interval}} = \frac{\max(\Delta\theta_t) - \min(\Delta\theta_t)}{I} \tag{15}$$

The $i$-th interval is then defined as:

$$\mathcal{I}i = [\min(\Delta\theta_t) + (i-1) \cdot \delta_{\text{interval}}, \min(\Delta\theta_t) + i \cdot \delta_{\text{interval}})], \quad \text{for } i = 1, \ldots, I \tag{16}$$

To perform group-wise pruning, we define a binary mask for each interval $I_i$, where pruning decisions are sampled from a Bernoulli distribution with success probability $1 - s_l$, and $s_l$ is the sparsity rate of the layer. Specifically:

$$\boldsymbol{M}_{I_i}^{(j,k)} \sim \begin{cases} \text{Bernoulli}(1 - s_l), & \text{if } {\Delta\theta_t}^{(j,k)} \in I_i \\ 0, & \text{otherwise} \end{cases} \tag{17}$$

The pruned delta weight ${\Delta\theta_t}^*$ is obtained by applying the group-wise Bernoulli masks to the quantized delta weights:

$$\Delta\theta_t^* = \sum_{i=1}^{I} \boldsymbol{M}_{\mathcal{I}_i} \odot \Delta\theta_t \tag{18}$$

## B  Detailed Derivation

### B.1  Derivation of the Variance-Entropy Relationship

We aim to derive the relationship between the variance and the entropy of a layer. First, we define the entropy of a random variable $L$ with probability density function $p(l)$ as:

$$H(L) = -\mathbb{E}[\log p(l)] \tag{19}$$

Assuming $L$ follows a Gaussian distribution $\mathcal{N}(\mu, \sigma^2)$, its probability density function is:

$$p(l) = \frac{1}{\sqrt{2\pi\sigma^2}} \exp\left(-\frac{(l-\mu)^2}{2\sigma^2}\right) \tag{20}$$

Taking the logarithm of the density function, we obtain:

$$\log p(l) = \log\left(\frac{1}{\sqrt{2\pi\sigma^2}} \exp\left(-\frac{(l-\mu)^2}{2\sigma^2}\right)\right) = -\frac{1}{2}\log(2\pi\sigma^2) - \frac{(l-\mu)^2}{2\sigma^2} \tag{21}$$

Substituting this into the definition of entropy, we have:

$$\begin{aligned} H(L) &= -\int_{-\infty}^{\infty} p(l)\left(-\frac{1}{2}\log(2\pi\sigma^2) - \frac{(l-\mu)^2}{2\sigma^2}\right) dl \\ &= \frac{1}{2}\log(2\pi\sigma^2)\int p(l)dl + \frac{1}{2\sigma^2}\int p(l)(l-\mu)^2 dl \end{aligned} \tag{22}$$

Next, we use known properties of the Gaussian distribution:

$$\int p(l)dl = 1 \quad \text{and} \quad \int p(l)(l-\mu)^2 dl = \text{Var}(L) = \sigma^2 \tag{23}$$

Substituting these into the expression for entropy yields:

$$\begin{aligned}
H(L) &= \frac{1}{2}\log(2\pi\sigma^2) + \frac{1}{2} \\
&= \frac{1}{2}\log(2\pi) + \log(\sigma) + \frac{1}{2}
\end{aligned} \tag{24}$$

Therefore, we can conclude that the entropy $H(L)$ increases logarithmically with the standard deviation $\sigma$ of the distribution. Since the variance is $\sigma^2$, entropy is monotonically increasing with variance. This derivation shows that the entropy of a layer is positively correlated with its variance. Since entropy represents the amount of information, layers with larger variance contain more information. As a result, such layers should be assigned a smaller sparsity rate to preserve more information.

### B.2 Derivation of the Variance of Activation Error

According to Eq. 13, the activation error $\epsilon$ is defined as:

$$\varepsilon = a \odot \left(1 - \frac{\gamma}{1-s}B\right), \quad \text{where } a = \Delta\theta \odot x \tag{25}$$

Here, $B$ is a Bernoulli random variable, where the probability of success is $1-s$. For a Bernoulli distribution, we know that the expectation and second moment are $\mathbb{E}[B] = 1-s$ and $\mathbb{E}[B^2] = 1-s$. We first calculate the expectation of $\epsilon$. Using the linearity of expectation:

$$\begin{aligned}
\mathbb{E}[\varepsilon] &= a \odot \mathbb{E}\left[1 - \frac{\gamma}{1-s}B\right] \\
&= a \odot \left(1 - \frac{\gamma}{1-s}\mathbb{E}[B]\right) \\
&= a \odot \left(1 - \frac{\gamma}{1-s}(1-s)\right) \\
&= a \odot (1-\gamma)
\end{aligned} \tag{26}$$

Next, we calculate the expectation of $\epsilon^2$:

$$\begin{aligned}
\mathbb{E}[\varepsilon^2] &= a^2 \odot \mathbb{E}\left[1 - 2\frac{\gamma}{1-s}B + \left(\frac{\gamma}{1-s}B\right)^2\right] \\
&= a^2 \odot \left[1 - 2\frac{\gamma}{1-s}\mathbb{E}[B] + \left(\frac{\gamma}{1-s}\right)^2\mathbb{E}[B]\right] \\
&= a^2 \odot \left[1 - 2\frac{\gamma}{1-s}(1-s) + \left(\frac{\gamma}{1-s}\right)^2(1-s)\right] \\
&= a^2 \odot \left(1 - 2\gamma + \frac{\gamma^2}{1-s}\right)
\end{aligned} \tag{27}$$

Now, we can calculate the variance of $\epsilon$. The variance is given by:

$$\begin{aligned}
\text{Var}(\varepsilon) &= \mathbb{E}[\varepsilon^2] - (\mathbb{E}[\varepsilon])^2 \\
&= a^2 \odot \left(1 - 2\gamma + \frac{\gamma^2}{1-s}\right) - a^2 \odot (1-\gamma)^2 \\
&= \boldsymbol{a}^2 \odot \left[\left(1 - 2\gamma + \frac{\gamma^2}{1-s}\right) - (1 - 2\gamma + \gamma^2)\right] \\
&= a^2 \odot \left(\frac{\gamma^2 s}{1-s}\right)
\end{aligned} \tag{28}$$

From the derived variance expression, we can observe that when the sparsity $s$ is high, the variance will become very large, leading to unstable performance. To mitigate this, we can choose a value for $\gamma$ in the range $[0, 1]$ to reduce the impact of the original rescaling factor. However, $\gamma$ cannot be chosen too small. This is because the expectation of the activation error is $a \odot (1 - \gamma)$, and if $\gamma$ is too small, the expectation of the activation error increases, which may lead to undesirable effects. Therefore, $\gamma$ is typically set in the range $[0.5, 1.0]$ under high sparsity, which balances both the variance and the expectation of the activation error.

### B.3 Detailed Derivation of Compression Ratio

**Golomb Coding.** Golomb coding [25] is a run-length entropy coding scheme particularly well-suited for a geometric distribution, where it encodes the distances between successive non-zero entries (i.e., run-lengths of zeros). Unlike conventional formats such as Compressed Sparse Row (CSR), which store explicit indices or pointers for non-zero values, run-length coding represents zero runs instead of storing indices, which can yield a more compact representation than CSR when sparsity is high.

**Compression Ratio Derivation.** In our setting, after pruning, most entries in the delta weight are zeros, and the positions of non-zeros can be modeled as a Bernoulli process with success probability equal to the density $1 - s$, where $s$ is the sparsity. Consequently, the run-lengths of zeros follow a geometric distribution, for which Golomb coding achieves an average code length close to the entropy of the underlying geometric distribution. The entropy of a geometric distribution is:

$$H_{\text{geo}} = -(1 - s) \log_2(1 - s) - s \log_2 s \tag{29}$$

When each non-zero entry can take $m$ discrete values, the maximum entropy contribution of its symbol is $\log_2 m$, which corresponds to the case where all $m$ values are equally likely. In practice, the values of our non-zero entries follow a truncated Gaussian distribution, so the actual entropy is smaller than $\log_2 m$, implying that the achievable compression ratio is higher.

For simplicity of analysis, we approximate this term by $\log_2 m$, which serves as an upper bound, and the per-parameter entropy of the compressed model is:

$$H_{\text{comp}} \approx -\left[ s \log_2 s + (1 - s) \log_2 \frac{(1-s)}{m} \right] \tag{30}$$

By contrast, the original uncompressed model stored in FP16 format requires $H_{\text{orig}} = 16$ bits per parameter. Therefore, the theoretical compression ratio is:

$$\text{CR} = \frac{H_{\text{orig}}}{H_{\text{comp}}} \approx \frac{16}{-\left[ s \log_2 s + (1 - s) \log_2 \frac{(1-s)}{m} \right]} \tag{31}$$

Note that the rescaling factor is represented by a single 16-bit value, which is negligible compared to the overall storage and is therefore ignored when reporting the compression ratio.

**Detailed Compression Ratios.** We also report the compression ratios and corresponding sparsity rates of DARE [91], MP [27], and UltraDelta under each setting. Practical compression ratios with Golomb coding are computed following Eq. 31, using $m = 16$ for DARE and MP (16-bit precision) and $m = 4$ for UltraDelta (4-bit quantization). For completeness, we also include Index-Free compression ratios, which denote the ideal upper bound where only non-zero values are stored. The results are shown in Tab. 8.

## C Additional Experimental Settings and Results

### C.1 HuggingFace IDs for LLM Checkpoints

We provide the Hugging Face IDs of the LLMs used in our experiments, as shown in Tab. 9. All models are evaluated using the lm-evaluation-harness [23] framework for general language tasks and EvalPlus [52] framework for code generation tasks.

Table 8: Compression ratios, average performance, and corresponding sparsity rates across different models. "Practical" denotes the practical compression ratio using Golomb coding, "Ideal" the ideal upper bound, and "Avg" the average performance.

| Method | LLaMA-2-7B (Tab. 1) | | | | LLaMA-2-13B (Tab. 1) | | | |
|---|---|---|---|---|---|---|---|---|
| | Practical | Ideal | Avg | Sparsity | Practical | Ideal | Avg | Sparsity |
| DARE [91] (Sparsity) | 14.7× | 20× | 38.79 | 95.0% | 23.7× | 33.3× | 45.63 | 97.0% |
| MP [27] (Sparsity) | 14.7× | 20× | 36.46 | 95.0% | 23.7× | 33.3× | 29.65 | 97.0% |
| DARE [91] (Compression) | 31.7× | 45.5× | 33.97 | 97.8% | 50.1× | 74.1× | 40.23 | 98.65% |
| MP [27] (Compression) | 31.7× | 45.5× | 27.02 | 97.8% | 50.1× | 74.1× | 15.36 | 98.65% |
| UltraDelta (Ours) | **32.9×** | **80.0×** | **45.57** | 95.0% | **50.9×** | **133.0×** | **52.05** | 97.0% |

| Method | Recent LLMs (Tab. 2) | | | | ViT-B/32 (Tab. 10) | | | |
|---|---|---|---|---|---|---|---|---|
| | Practical | Ideal | Avg | Sparsity | Practical | Ideal | Avg | Sparsity |
| DARE [91] (Sparsity) | 14.7× | 20× | 49.18 | 95.0% | 23.7× | 33.3× | 89.7 | 97.0% |
| MP [27] (Sparsity) | 14.7× | 20× | 41.68 | 95.0% | 23.7× | 33.3× | 77.4 | 97.0% |
| DARE [91] (Compression) | – | – | – | – | 50.1× | 74.1× | 83.5 | 98.65% |
| MP [27] (Compression) | – | – | – | – | 50.1× | 74.1× | 53.1 | 98.65% |
| UltraDelta (Ours) | **32.9×** | **80.0×** | **52.05** | 95.0% | **50.9×** | **133.0×** | **90.7** | 97.0% |

| Method | ViT-L/14 (Tab. 4) | | | | T5-base (Tab. 3) | | | |
|---|---|---|---|---|---|---|---|---|
| | Practical | Ideal | Avg | Sparsity | Practical | Ideal | Avg | Sparsity |
| DARE [91] (Sparsity) | 66.5× | 100.0× | 93.6 | 99.0% | 127.6× | 200.0× | 85.90 | 99.5% |
| MP [27] (Sparsity) | 66.5× | 100.0× | 59.7 | 99.0% | 127.6× | 200.0× | 63.97 | 99.5% |
| DARE [91] (Compression) | 127.6× | 200.0× | 89.1 | 99.5% | 220.5× | 357.0× | 84.30 | 99.72% |
| MP [27] (Compression) | 127.6× | 200.0× | 29.5 | 99.5% | 220.5× | 357.0× | 60.44 | 99.72% |
| UltraDelta (Ours) | **132.5×** | **400.0×** | **94.4** | 99.0% | **224.6×** | **800.0×** | **86.74** | 99.5% |

| Method | RoBERTa-base (Tab. 11) | | | | BEiT-3 (Tab. 5) | | | |
|---|---|---|---|---|---|---|---|---|
| | Practical | Ideal | Avg | Sparsity | Practical | Ideal | Avg | Sparsity |
| DARE [91] (Sparsity) | 14.7× | 20.0× | 83.23 | 95.0% | 7.7× | 10.0× | 70.74 | 90.0% |
| MP [27] (Sparsity) | 14.7× | 20.0× | 80.06 | 95.0% | 7.7× | 10.0× | 16.59 | 90.0% |
| DARE [91] (Compression) | 31.7× | 45.5× | 73.12 | 97.8% | 18.1× | 25.0× | 54.49 | 96.0% |
| MP [27] (Compression) | 31.7× | 45.5× | 69.87 | 97.8% | 18.1× | 25.0× | 16.47 | 96.0% |
| UltraDelta (Ours) | **32.9×** | **80.0×** | **84.46** | 95.0% | **18.4×** | **40.0×** | **73.95** | 90.0% |

## C.2 Experiments on eight ViT-B/32 models

**Results.** The results of ViT-B/32 models are shown in Tab. 10. With 4-bit quantization and 97% sparsity, UltraDelta achieves 50.9× compression ratio on ViT-B/32. UltraDelta achieves an average accuracy of 90.7 across 8 diverse vision tasks, nearly matching the performance of individually fine-tuned models (91.0). Compared to all baselines, UltraDelta achieves both higher compression and better accuracy.

## C.3 Experiments on eight RoBERTa-base models

**Settings.** For RoBERTa-base, we report Matthews correlation on CoLA, the average of Pearson and Spearman correlations on STS-B, and accuracy on the other tasks.

**Results.** The results of RoBERTa-base models are shown in Tab. 11. For RoBERTa, we apply 4-bit quantization and prune 95% of parameters, achieving an overall 32.9× compression ratio. Despite the high compression, UltraDelta achieves an average score of 84.46, closely matching the uncompressed individually fine-tuned model (85.56) and outperforming all baselines by a notable margin.

Table 9: LLM checkpoint IDs.

| Models | Pre-trained | Fine-tuned |
|---|---|---|
| **Model Size 7B (Llama-2 Series)** | | |
| Chat | *meta-llama/Llama-2-7b-hf* | *meta-llama/Llama-2-7b-chat-hf* |
| Code | *codellama/CodeLlama-7b-Python-hf* | *vanillaOVO/WizardCoder-Python-7B-V1.0* |
| Math | *meta-llama/Llama-2-7b-hf* | *WizardLMTeam/WizardMath-7B-V1.0* |
| **Model Size 13B (Llama-2 Series)** | | |
| Chat | *meta-llama/Llama-2-13b-hf* | *meta-llama/Llama-2-13b-chat-hf* |
| Code | *codellama/CodeLlama-13b-Python-hf* | *WizardLMTeam/WizardCoder-Python-13B-V1.0* |
| Math | *meta-llama/Llama-2-13b-hf* | *vanillaOVO/WizardMath-13B-V1.0* |
| **Recent Models** | | |
| Llama-3.1 | *meta-llama/Llama-3.1-8B* | *allenai/Llama-3.1-Tulu-3-8B-SFT* |
| Qwen2.5 | *Qwen/Qwen2.5-7B* | *Qwen/Qwen2.5-7B-Instruct* |
| Qwen3 | *Qwen/Qwen3-8B* | *Qwen/Qwen3Guard-Gen-8B* |

Table 10: Performance comparison on ViT-B/32 models across 8 tasks.

| Methods | Ratio | SUN397 | Cars | RESISC45 | EuroSAT | SVHN | GTSRB | MNIST | DTD | Avg |
|---|---|---|---|---|---|---|---|---|---|---|
| Individual | 1× | 79.2 | 77.7 | 96.1 | 99.8 | 97.4 | 98.7 | 99.7 | 79.4 | 91.0 |
| Traditional MTL [8] | 8× | 73.9 | 74.4 | 93.9 | 98.2 | 95.8 | 98.9 | 99.5 | 77.9 | 89.1 |
| BitDelta [53] | 16× | 78.3 | 75.9 | 95.4 | 99.3 | 96.4 | 98.2 | 99.2 | 78.4 | 90.1 |
| Delta-CoMe [64] | 16× | 78.5 | 72.3 | 95.6 | 99.6 | **97.3** | 98.6 | 99.5 | 78.3 | 90.0 |
| DARE [91] (Sparsity) | 23.7× | 76.0 | 73.3 | 95.1 | **99.7** | 97.0 | 98.5 | 99.6 | 78.1 | 89.7 |
| MP [27] (Sparsity) | 23.7× | 58.1 | 44.2 | 91.2 | 96.1 | 91.3 | 73.5 | 98.3 | 66.7 | 77.4 |
| DARE [91] (Compression) | 50.1× | 57.2 | 55.2 | 91.3 | 99.4 | 96.4 | 98.2 | 99.6 | 71.1 | 83.5 |
| MP [27] (Compression) | 50.1× | 18.8 | 16.4 | 81.8 | 88.3 | 70.3 | 28.5 | 77.8 | 42.7 | 53.1 |
| **UltraDelta (Ours)** | **50.9×** | **78.6** | **77.4** | **95.7** | 99.4 | 97.0 | **98.7** | **99.7** | **79.0** | **90.7** |

## C.4 Detailed Results of Ablation Study

### C.4.1 Ablation on Distribution-Aware Compression

We present the detailed numerical results of the ablation study conducted on DAC. For brevity, we report only the results under 90% sparsity, as summarized in Tab. 12. Following [32, 93], we use ViT-B/16 as the pre-trained backbone model and evaluate its performance across 30 diverse image classification datasets. Specifically, the datasets are: MNIST [44], CIFAR-10 [41], CIFAR-100 [41], Cars [40], Fashion-MNIST [85], EMNIST [16], STL10 [14], GTSRB [73], SHVN [92], Oxford-IIIT Pet [62], Cats and Dogs [18], Dogs [39], Beans [42], Food-101 [7], Fruits-360 [59], Vegetables [1], MangoLeafBD [2], Flowers Recognition [58], Landscape Recognition [19], Weather [86], DTD [13], EuroSAT [29], RESISC45 [12], SUN397 [87], KenyanFood13 [37], Intel Images [5], Garbage Classification [9], Animal-10N [72], CUB-200-2011 [80], and Kvasir-v2 [65]. These datasets are chosen to comprehensively evaluate generalization across various data distributions and classification challenges. Accuracy is used as the evaluation metric for all datasets.

Table 11: Performance comparison on RoBERTa-base models across 8 tasks.

| Methods | Ratio | CoLA | SST2 | MRPC | STSB | QQP | MNLI | QNLI | RTE | Avg |
|---|---|---|---|---|---|---|---|---|---|---|
| Individual | 1× | 60.18 | 94.04 | 89.22 | 90.63 | 91.41 | 87.20 | 92.71 | 79.06 | 85.56 |
| BitDelta [53] | 16× | 36.83 | 93.12 | 88.73 | 84.82 | **90.00** | **85.57** | 91.73 | 70.40 | 80.15 |
| Delta-CoMe [64] | 16× | 57.67 | 92.48 | 87.99 | 90.53 | 89.17 | 82.66 | **92.57** | 76.9 | 83.74 |
| DARE [91] (Sparsity) | 14.7× | 59.30 | 93.92 | 88.97 | 90.53 | 86.55 | 77.84 | 92.13 | 76.53 | 83.23 |
| MP [27] (Sparsity) | 14.7× | 57.69 | 92.43 | 84.31 | 88.18 | 86.26 | 74.40 | 86.80 | 70.40 | 80.06 |
| DARE [91] (Compression) | 31.7× | 58.22 | 93.69 | 87.50 | 90.27 | 53.47 | 36.48 | 89.51 | 75.81 | 73.12 |
| MP [27] (Compression) | 31.7× | 51.89 | 91.63 | 78.68 | 86.44 | 65.77 | 53.49 | 65.64 | 65.43 | 69.87 |
| **UltraDelta (Ours)** | **32.9×** | **62.64** | **94.38** | **89.22** | **90.60** | 85.79 | 82.92 | 92.15 | **77.98** | **84.46** |

Table 12: Performance of DAC and DARE on ViT-B/16 models across 30 tasks. "(w/ $b$-bit)" denotes the quantized setting using $b$-bit precision, while "(w/o q.)" refers to the non-quantized setting.

| Methods | Animal-10N | Beans | Cats and Dogs | Cifar-10 | Cifar-100 | CUB-200-2011 | Dogs | DTD | EMNIST | EuroSAT |
|---|---|---|---|---|---|---|---|---|---|---|
| Individual | 92.46 | 97.70 | 99.05 | 97.88 | 89.85 | 84.78 | 89.91 | 81.1 | 94.67 | 99.07 |
| DARE [53] (w/ 4-bit) | 92.26 | 34.87 | 98.17 | 33.39 | 53.37 | 84.28 | 89.55 | 81.08 | 94.69 | 99.07 |
| **UltraDelta (w/ 4-bit) (Ours)** | 92.42 | 97.32 | 98.98 | 58.33 | 72.14 | 83.98 | 89.74 | 80.82 | 94.51 | 99.04 |
| DARE [53] (w/ 8-bit) | 92.26 | 34.87 | 98.17 | 33.39 | 53.37 | 84.28 | 89.55 | 81.08 | 94.69 | 99.07 |
| **UltraDelta (w/ 8-bit) (Ours)** | 92.68 | 98.47 | 98.66 | 80.06 | 78.12 | 84.54 | 90.18 | 81.13 | 94.61 | 99.00 |
| DARE [53] (w/o q.) | 92.28 | 97.70 | 98.04 | 57.90 | 56.07 | 84.19 | 89.91 | 81.15 | 94.58 | 99.15 |
| **UltraDelta (w/o q.) (Ours)** | 92.30 | 98.08 | 98.99 | 96.77 | 80.83 | 84.28 | 90.00 | 81.08 | 94.62 | 99.07 |

| | Fashion | Flowers | KenyanFood13 | Food-101 | Fruits-360 | Garbage | GTSRB | Intel-Images | Kvasir-v2 | LandScape |
|---|---|---|---|---|---|---|---|---|---|---|
| Individual | 93.26 | 98.19 | 82.58 | 87.87 | 99.64 | 98.58 | 95.74 | 94.87 | 93.91 | 94.00 |
| DARE [53] (w/ 4-bit) | 84.09 | 98.19 | 82.70 | 38.00 | 99.64 | 98.61 | 7.57 | 93.33 | 74.81 | 92.40 |
| **UltraDelta (w/ 4-bit) (Ours)** | 79.58 | 98.17 | 82.21 | 29.14 | 99.58 | 98.54 | 87.16 | 90.93 | 56.47 | 93.80 |
| DARE [53] (w/ 8-bit) | 84.09 | 98.19 | 82.70 | 38.00 | 99.64 | 98.61 | 75.70 | 93.33 | 74.81 | 92.40 |
| **UltraDelta (w/ 8-bit) (Ours)** | 83.85 | 98.19 | 82.45 | 47.03 | 99.59 | 98.50 | 91.96 | 93.93 | 89.94 | 73.80 |
| DARE [53] (w/o q.) | 73.69 | 98.17 | 82.45 | 43.93 | 99.64 | 98.61 | 62.49 | 92.13 | 77.62 | 93.40 |
| **UltraDelta (w/o q.) (Ours)** | 90.10 | 98.12 | 82.82 | 61.87 | 99.63 | 98.50 | 95.36 | 94.13 | 92.22 | 94.20 |

| | MangoLeafBD | MNIST | Pet | RESISC45 | Cars | STL10 | SUN397 | SVHN | Vegetables | Weather |
|---|---|---|---|---|---|---|---|---|---|---|
| Individual | 100.00 | 99.22 | 92.20 | 99.00 | 85.29 | 99.08 | 87.50 | 96.22 | 100.00 | 98.19 |
| DARE [53] (w/ 4-bit) | 98.47 | 78.59 | 91.88 | 98.90 | 79.78 | 99.09 | 85.24 | 96.22 | 91.67 | 97.62 |
| **UltraDelta (w/ 4-bit) (Ours)** | 85.95 | 98.93 | 92.56 | 98.95 | 79.74 | 99.21 | 85.82 | 96.21 | 91.30 | 80.46 |
| DARE [53] (w/ 8-bit) | 98.47 | 78.59 | 91.88 | 98.90 | 79.78 | 99.09 | 85.24 | 96.22 | 91.67 | 97.62 |
| **UltraDelta (w/ 8-bit) (Ours)** | 100.00 | 98.68 | 92.37 | 99.10 | 80.11 | 99.16 | 85.74 | 96.33 | 99.67 | 97.51 |
| DARE [53] (w/o q.) | 100.00 | 64.62 | 92.26 | 98.79 | 79.77 | 99.09 | 85.55 | 96.30 | 99.83 | 87.03 |
| **UltraDelta (w/o q.) (Ours)** | 100.00 | 98.97 | 92.64 | 99.13 | 80.55 | 99.22 | 85.90 | 96.32 | 99.97 | 97.90 |

| **Average Acc** | Individual | DARE (w/ 4-bit) | **UltraDelta (w/ 4-bit) (Ours)** | DARE (w/ 8-bit) | **UltraDelta (w/ 8-bit) (Ours)** | DARE (w/o q.) | **UltraDelta (w/o q.) (Ours)** |
|---|---|---|---|---|---|---|---|
| Acc | 94.06 | 81.58 | **86.40** | 83.85 | **90.18** | 85.88 | **92.45** |

### C.4.2 Ablation on Hyper-Parameters

We present the detailed numerical results of our hyper-parameter ablation studies in Tab. 13. The results complement the analysis provided in the main text and offer full visibility into the performance under different settings.

Table 13: Performance of different hyperparameter settings applied to ViT-B/32.

| Method | SUN397 | Cars | RESISC45 | EuroSAT | SVHN | GTSRB | MNIST | DTD | Avg. |
|---|---|---|---|---|---|---|---|---|---|
| **Distribution-Aware Compression (quantization bit)** | | | | | | | | | |
| 2-bit | 73.4 | 70.1 | 94.4 | 99.1 | 96.9 | 98.2 | 99.7 | 74.3 | 88.3 |
| 4-bit | 78.6 | 77.4 | 95.7 | 99.4 | 97.0 | 98.7 | 99.7 | 79.0 | 90.7 |
| 6-bit | 78.7 | 77.5 | 95.7 | 99.5 | 97.1 | 98.8 | 99.7 | 78.6 | 90.7 |
| 8-bit | 78.8 | 77.7 | 95.7 | 99.7 | 97.0 | 98.7 | 99.7 | 79.0 | 90.8 |
| **Variance-Based Mixed Sparsity Allocation (sparsity step size)** | | | | | | | | | |
| Step = 0.000 | 77.3 | 75.1 | 95.5 | 99.7 | 97.1 | 98.8 | 99.6 | 77.6 | 90.1 |
| Step = 0.005 | 77.9 | 75.7 | 95.4 | 99.7 | 97.0 | 98.8 | 99.7 | 78.9 | 90.4 |
| Step = 0.010 | 78.4 | 76.7 | 95.6 | 99.4 | 97.0 | 98.9 | 99.6 | 78.9 | 90.6 |
| Step = 0.020 | 78.6 | 77.4 | 95.7 | 99.4 | 97.0 | 98.7 | 99.7 | 79.0 | 90.7 |
| Step = 0.025 | 78.3 | 76.4 | 95.1 | 98.1 | 96.6 | 98.6 | 99.7 | 79.2 | 90.3 |
| Step = 0.030 | 78.5 | 75.3 | 94.5 | 96.5 | 96.7 | 98.2 | 99.7 | 78.7 | 89.8 |

# D  Discussion

## D.1  Regularization Effect

We observe in experiments that UltraDelta sometimes even outperforms fine-tuned models, especially in the LLM setting. This can be attributed to overfitting or underfitting during fine-tuning.

To verify this, we conducted controlled experiments on two datasets, SUN397 [87] and Cars [40], using a pretrained ViT-B/32 model. By varying the number of fine-tuning steps, we controlled the degree of model fitting. The results are shown in Tab. 14, and we find that underfitted models (early training steps, higher loss) consistently benefit from UltraDelta, while well-trained models (later steps, lower loss) show some or no improvement. These results indicate that UltraDelta mainly benefits underfitted models, while well-trained models already capture most task-specific information and thus leave little room for further improvement.

This also explains why improvements are more common in LLMs, which are typically evaluated in zero-shot or few-shot settings. In such scenarios, these models are not explicitly fine-tuned on the target tasks, which often leads to weaker generalization. As a result, they can be regarded as underfitted to the evaluation benchmarks. In this context, UltraDelta provides a regularization effect, enabling underfitted models to benefit more from compression.

Table 14: Controlled experiments on ViT-B/32 with varying training steps. "UltraDelta Accuracy" denotes the test accuracy after applying UltraDelta.

| Training Steps | 1000 | 2000 | 3000 | 4000 | 5000 | 6000 | 7000 | 8000 | 9000 | 10000 |
|---|---|---|---|---|---|---|---|---|---|---|
| **Cars [40]** | | | | | | | | | | |
| Training Loss | 0.1404 | 0.0834 | 0.0206 | 0.0069 | 0.0059 | 0.0059 | 0.0034 | 0.0077 | 0.0048 | 0.0036 |
| Test Accuracy | 72.86 | 68.90 | 75.22 | **78.78** | **77.74** | **78.37** | **78.01** | **75.21** | **77.60** | **77.34** |
| **UltraDelta Accuracy** | **74.02** | **73.27** | **75.82** | 75.56 | 75.62 | 75.66 | 76.13 | 72.80 | 74.74 | 74.06 |
| **SUN397 [87]** | | | | | | | | | | |
| Training Loss | 0.5475 | 0.1432 | 0.0426 | 0.1125 | 0.0526 | 0.0040 | 0.0018 | 0.0009 | 0.0006 | 0.0004 |
| Test Accuracy | 73.63 | 75.94 | 75.40 | 74.63 | 74.07 | **76.94** | 76.87 | 77.03 | 77.06 | 77.07 |
| **UltraDelta Accuracy** | **76.02** | **76.92** | **77.21** | **76.20** | **76.19** | 76.66 | **76.94** | **77.11** | **77.19** | **77.31** |

## D.2 Compression Ratio Differences Across Models

We observe that different models can be compressed to different extents without losing performance. This variation mainly comes from the amount of redundancy in their delta weights. For instance, T5-base has relatively high redundancy, making it possible to compress it up to $224\times$ while maintaining or even improving performance. In contrast, models like BEiT-3 have delta weights with larger magnitudes and lower redundancy, leaving less room for aggressive compression. This suggests that the achievable compression ratio depends not only on the compression method itself, but also on the model architecture, its training process, and the nature of the fine-tuned tasks.

## D.3 Limitations and Future Work

**Heuristic Rescaling Factor.** One limitation of our method lies in the heuristic nature of the additional rescaling factor $\gamma$. We set $\gamma$ according to the trace norm of each delta weight, which serves as a lightweight proxy for activation instability under the data-free setting. While this simple strategy helps mitigate instability at high sparsity, it remains a heuristic and may not be optimal. Exploring more principled or adaptive approaches for determining $\gamma$ under data-free constraints remains an important direction for future work.

**Deployment Efficiency.** Another limitation is that our method is not designed to directly optimize deployment efficiency, such as inference acceleration. Achieving real speedup typically requires specialized GPU kernels or structured sparsity to better exploit hardware parallelism. Future work may combine our approach with optimized kernels or structured pruning to realize both strong compression–performance trade-offs and practical deployment gains.

