# OpenReview forum: "Breaking the Compression Ceiling: Data-Free Pipeline for Ultra-Efficient Delta Compression"
_NeurIPS.cc/2025/Conference — NeurIPS 2025 poster_

### Official Review · Reviewer_8YfX · 2025-06-28

**Clarity:** 3
**Significance:** 2
**Originality:** 1
**Rating:** 2
**Confidence:** 4

**Summary:**

This paper proposes UltraDelta, a new data-free compression pipeline designed to reduce the storage overhead of fine-tuned models by aggressively compressing delta weights (the difference between a fine-tuned model and its base pretrained model).

**Questions:**

See limitations

**Ethical Concerns:**

["NO or VERY MINOR ethics concerns only"]

**Final Justification:**

------- Question 1 : Limited Novelty ----------

ComPEFT [4] can serve for Full Finetuning Updates, in its section 5.3, it demonstrate the effectiveness in compressing Full Finetuning Updates, the author should carefully read the references and give a comprehensive discussion, again, combining
ComPEFT [4] can indeed be data-free, as state in introduction "ComPEFT retains high performance post-hoc compression without the need for additional training"
ComPEFT [4] contain two steps: pruning and quantization which is followed by the paper closely.
As a result, without proper reference to related work is completely wrong.

------- Question 2 : Weak Theoretical Foundation------

DAREx[3] not only consider weight but also the activations, where its c represents $\Delta Wx$, so they are more comprehensive in analyzing the output change.

Your response on "our analysis is based on weight variance, which is fully data-free" is wrong, your $a= \Delta \theta x$ which rely on the input $x$, so your analysis is not data-free.....

Finally, it is the variance being the key points rather than the entropy, you Theoretical Analysis of the Rescaling Factor eventually analyze the activation error $\epsilon$ rather than entropy as state in line 185, this is the same as DAREx[3]'s theorem 3.1, you can completely remove the entropy term but just bound the output change as a direct signal on performance.

As a result, without proper reference to related work is wrong.

-------------- Question 4 : Unclear Rescaling Factor Determination--------

As shown in above response, the author wrongly believe their analysis is "data-free" and it rely on the assumption of Gaussian Distribution of weight and "ignoring" the influence of input $x$, thus the rescaling depend on weight is not comprehensive due to ignoring the input $x$ and may perform bad when $x$ activation is large (for example massive activation[5] in LLM)

[5] Massive Activations in Large Language Models

Due to the wrong statement and reference on related work for novelty, lack reference to prior theory analysis, lack understanding on their own theory analysis, I think this paper should be rejected.

**Limitations:**

1. Limited Novelty: The core ideas—layer-wise importance for adaptive pruning [1], delta parameter quantization [2], and rescaling factor adjustments [3]—are each incremental extensions of established compression and pruning techniques. Furthermore, their combination has already been studied in works such as ComPEFT [4], which uses sparsification and ternary quantization for delta compression. This paper does not provide novel methodology beyond re-combining existing ideas, nor does it adequately acknowledge or cite these prior works (for example ComPEFT [4]).

2. Weak Theoretical Foundation: The theoretical rationale, such as the variance–entropy relationship, is presented in an overly simplistic manner without rigorous proofs or formal guarantees. What's more, [3] already provides a comprehensive theoretical treatment of variance’s effect on pruning error; the current paper should reference and build on that work directly.

3. The discussion in related work is not proper that methods like DAREx [3] and BitDelta [2] can operate with only small amounts of unlabeled data in an training-free unsupervised fashion, providing practical data efficiency and easy to implement, so this is not a disadvantage.

4. Unclear Rescaling Factor Determination: The trace-norm-based rescaling factor γ is claimed to be inversely proportional to the trace norm, but it is not clear why it is not simply set equal to its inverse. As discussed in lines 549–552 in appendix, the factor appears manually chosen within a heuristic range. A more detailed derivation or empirical justification is necessary to clarify how γ should be set in practice.

**Quality:**

2

**Strengths And Weaknesses:**

Strengths:

1. The paper addresses an important problem in delta parameter compression for large-scale fine-tuned models.

2. It includes an extensive empirical evaluation covering multiple model families and tasks, demonstrating broad experimental effort.

3. The manuscript is generally well-structured and easy to follow.

Limitations:

1. Limited Novelty: The core ideas—layer-wise importance for adaptive pruning [1], delta parameter quantization [2], and rescaling factor adjustments [3]—are each incremental extensions of established compression and pruning techniques. Furthermore, their combination has already been studied in works such as ComPEFT [4], which uses sparsification and ternary quantization for delta compression. This paper does not provide novel methodology beyond re-combining existing ideas, nor does it adequately acknowledge or cite prior works (for example ComPEFT [4]).

2. Weak Theoretical Foundation: The theoretical rationale, such as the variance–entropy relationship, is presented in an overly simplistic manner without rigorous proofs or formal guarantees. What's more, [3] already provides a comprehensive theoretical treatment of variance’s effect on pruning error; the current paper should reference and build on that work directly.

3. The discussion in related work is not proper that methods like DAREx [3] and BitDelta [2] can operate with only small amounts of unlabeled data in an training-free unsupervised fashion, providing practical data efficiency and easy to implement, so this is not a disadvantage.

4. Unclear Rescaling Factor Determination: The trace-norm-based rescaling factor γ is claimed to be inversely proportional to the trace norm, but it is not clear why it is not simply set equal to its inverse. As discussed in lines 549–552 in appendix, the factor appears manually chosen within a heuristic range. A more detailed derivation or empirical justification is necessary to clarify how γ should be set in practice.

[1] LAYER-ADAPTIVE SPARSITY FOR THE MAGNITUDE-BASED PRUNING

[2] BitDelta: Your Fine-Tune May Only Be Worth One Bit

[3] DARE the Extreme: Revisiting Delta-Parameter Pruning For Fine-Tuned Models

[4] ComPEFT: Compression for Communicating Parameter Efficient Updates via Sparsification and Quantization

---

> ### Author Rebuttal · Authors · 2025-07-29
>
> We sincerely thank the reviewer for the positive feedback, including: **(i) addressing an important problem for large-scale fine-tuned models; (ii) extensive experiments; (iii) well-structured and easy to follow**. We now address the reviewer’s questions in detail below.
>
> ---
>
> # Question 1 : Limited Novelty
>
> We appreciate the reviewer’s comment. Our novelty lies in **the design, motivation, and strict data-free constraint**.
>
> **1. The first data-free pipeline for ultra-efficient delta compression** :
>
> To the best of our knowledge, we are **the first to propose a data-free pipeline for ultra-efficient delta compression**. This makes our approach **practical in scenarios where tuning or data is unavailable**. Our method maintains **strong performance** under strict data-free constraints, achieving **high compression ratios** without additional data or tuning.
>
> **2. Novel motivation and design** :
>
> Our method is guided by a clear and novel motivation: **maximizing information preservation**. All components are **specifically designed** around this motivation.
>
> - **Variance-based mix sparsity allocation** uses **weight variance** as a simple, data-free proxy for **information content**, theoretically supported by the variance–entropy relation and rate–distortion theory.
>
> - **Distribution-aware compression** preserves the relative proportions of quantized values to **better preserve the intra-layer information**. This distribution-aware compression, to our knowledge, is not used in prior delta compression work.
>
> - **Trace-norm-guided rescaling** is designed to estimate an additional rescaling factor using trace norm to stabilize the pruned model based on trace norm, while remaining **fully data-free**.
>
> While we understand the reviewer’s concern, we believe it is not appropriate to negate our contributions by **referring to broad topics such as layer-wise pruning or delta compression quantization**. These are widely studied areas, and many works explore them with different goals and techniques. Our method introduces **novel designs and motivation**, which we believe **provide novelty within these broader topics**.
>
> **3. Clarifications on related works** :
>
> - LAMP[1] proposes an importance score **based on weight magnitude**, whereas **we design a variance-based score** from an infomation-theoretic perspective. The metric we use for layer-wise sparsity is quite different.
> - BitDelta[2] applies **binary masks and scaling factors** to quantize delta weights. This approach is **fundamentally different from our distribution-aware compression**, which uses uniform quantization combined with group-wise pruning to preserve the relative proportions of quantized values.
> - DAREx[3] **relies on data (labeled validation or unlabeled)**, and its own results show that using unlabeled data generally underperforms labeled validation data. However, using labeled validation data may cause the **risk of performance inflation**, which our approach avoids entirely.
> - ComPEFT[4] applies **magnitude pruning and quantizes the remaining weights with a fixed shared value**. The value is determined using validation data. It is **not data-free**, it does **not implement rescaling factor adjustment**, and its **sparsity and quantization strategies differ from ours**. Moreover, ComPEFT[4] is mainly designed for **PEFT models**, making its application scope different from UltraDelta, which targets general delta compression across full fine-tuned models.
> - We have already cited [1], [2], and [3] in our paper, and we will include a citation of ComPEFT[4] in the revised version.
>
> ---
>
> # Question 2 : Weak Theoretical Foundation
>
> 1. We thank the reviewer for the comments on our theoretical analysis. While we understand the concerns raised here, we would like to point out that other reviewers had a different perspective. **Reviewer vAk5** mentioned that *“the paper provides good theoretical motivation and empirical validation”*, and **Reviewer 9VE5** noted that *“it’s great to see the authors provide not only empirical results, but also theoretically analyze the effect of MSA.”*
>
> 2. To further strengthen this part, we have included an additional theoretical analysis based on **rate-distortion theory** (thanks to Reviewer 9VE5). Let the original weights be $W\\sim\\mathcal{N}(\\mu,\\sigma^2)$, and the compressed weights be $W^′$. The distortion is defined as the mean squared error (MSE) between them:
>    $$
>    D=\\mathbb{E}[(W-W’)^2]
>    $$
> According to classical rate-distortion theory for a Gaussian source, the minimum number of bits (information rate) required to represent $W$ under distortion $D$ is:
> $$
> R(D)=
> \\begin{cases}
> \\frac{1}{2}\\log\\frac{\\sigma^2}{D}, & 0 \\leq D \\leq \\sigma^2 \\\\
> 0, & D > \\sigma^2
> \\end{cases}
> $$
> Here, $R(D)$ measures the information needed to encode the weights while maintaining a reconstruction error no larger than $D$. This relationship shows that, for a fixed distortion $D$, weight with **larger variance $\\sigma^2$ require more information to represent**. Therefore, **layers with high weight variance are more “information-rich”** and should be preserved more carefully during pruning to maximize information preservation.
>
> 3. Regarding DAREx[3], we would like to clarify that the variance discussed in DAREx[3] (Theorem 3.1) is not the same as ours. The variance in DAREx[3] refers to **activation variance**, which requires data to obtain activations. In contrast, our analysis is based on **weight variance**, which is fully data-free. Additionally, DAREx[3] studies activation variance for the purpose of **rescaling factor adjustment**, our use of weight variance is for **layer-wise sparsity allocation**. The two approaches **differ in the definition of variance, data requirements, and intended use**.
>
> ---
>
> # Question 3 : Data Usage in BitDelta[2] and DAREx[3]
>
> 1. Compared with DAREx[3] and BitDelta[2], our method does **not require any labeled or unlabeled data**, nor does it involve additional tuning or retraining. This makes UltraDelta **highly generalizable**, saves computation and deployment time, and is practical for real-world applications where data availability is limited or privacy concerns exist.
>
> 2. Although DAREx[3] can use unlabeled data, its experiments show that tuning with unlabeled data performs worse than using labeled validation data, while the latter **risks evaluation inflation**.
>
> 3. BitDelta[2] uses 800 samples (length 128) from the C4 dataset for calibration, introducing **data and time dependencies** that limit its applicability in data-restricted environments. We also report the compression runtime comparison between BitDelta[2] and UltraDelta:
>
>    | Method  | BitDelta | UltraDelta |
>    | ------- | -------- | ---------- |
>    | Runtime | 326.96s  | **211.53s**    |
>
> ---
>
> # Question 4 : Unclear Rescaling Factor Determination
>
> 1. The trace norm measures **the overall magnitude and effective rank of the delta weight**, reflecting how much information it carries. A lower trace norm typically indicates lower effective rank and less information.
>
> 2. Motivated by **information preservation**, we use the trace norm as **a simple and data-free proxy** for information. We also empirically validate that the optimal rescaling factor is inversely related to the trace norm (see **Figure 3**).
>
> 3. Although this is a heuristic, it is **data-free, theoretically motivated, and empirically validated** in our experiments. We hope it can open **an interesting direction for future work** on data-free rescaling factor adjustment.
>
> ---
> [1] LAYER-ADAPTIVE SPARSITY FOR THE MAGNITUDE-BASED PRUNING
>
> [2] BitDelta: Your Fine-Tune May Only Be Worth One Bit
>
> [3] DARE the Extreme: Revisiting Delta-Parameter Pruning For Fine-Tuned Models
>
> [4] ComPEFT: Compression for Communicating Parameter Efficient Updates via Sparsification and Quantization

---

> ### Comment · Reviewer_8YfX · 2025-08-05
>
> ------- Question 1 : Limited Novelty ----------
>
>  1. ComPEFT [4]  **can serve for Full Finetuning Updates**, in its section 5.3, it demonstrate the effectiveness in compressing Full Finetuning Updates, the author should carefully read the references and give a comprehensive discussion, again,  combining
> 2. ComPEFT [4] **can indeed be data-free**, as state in introduction "ComPEFT retains high performance post-hoc compression without the need for additional training"
> 3. ComPEFT [4]  contain two steps: **pruning and quantization** which is followed by the paper closely.
>
> As a result, without proper reference to related work is completely wrong.
>
> ------- Question 2 : Weak Theoretical Foundation------
> 1. DAREx[3]  not only consider weight but also the activations, where its c represents $\Delta Wx$, so they are more comprehensive in analyzing the output change.
>
> 2. Your response on "our analysis is based on weight variance, which is fully data-free" is wrong, your $a= \Delta \theta x$ which rely on the input $x$, so your analysis is not data-free.....
>
> 3. Finally, it is the variance being the key points rather than the entropy, you Theoretical Analysis of the Rescaling Factor **eventually analyze the activation error $\epsilon$ rather than entropy as state in line 185**, this is the same as DAREx[3]'s theorem 3.1, you can completely remove the entropy term but just  bound the output change as a direct signal on performance.
>
> As a result, without proper reference to related work is wrong.
>
> -------------- Question 4 : Unclear Rescaling Factor Determination--------
>
> As shown in above response, the author wrongly believe their analysis is  "data-free" and it rely on the assumption of Gaussian Distribution of weight and "ignoring" the influence of input $x$, thus the rescaling depend on weight is not comprehensive due to ignoring the input $x$ and may perform bad when $x$ activation is large (for example massive activation[5] in LLM)
>
> [5] Massive Activations in Large Language Models
>
>
>
> Due to the wrong statement and reference on related work for novelty, lack reference to prior theory analysis, lack understanding on their own theory analysis, I think this paper should be rejected.

---

> > ### Author Response · Authors · 2025-08-05
> >
> > # Question 1:
> > We thank the reviewer for pointing out ComPEFT. We clarify as follows:
> >
> > 1. As we mentioned in the rebuttal, "*ComPEFT is **mainly** designed for PEFT models*". Its abstract explicitly states: “*To address these issues, we present ComPEFT, a novel method for compressing fine-tuning residuals (task vectors) of **PEFT-based models**.*” This indicates **its primary scope is PEFT-based models** rather than FFT deltas.
> >
> > 2. The phrase “*without the need for **additional training***” only means that **ComPEFT does not require further training**. **It does not imply being fully data-free**: in Sec. 3.2, its quantization method specifies “*The value of $\\alpha$ is selected by evaluating a metric of choice on **a small validation set**.*”
> >
> > 3. We also clarified in the rebuttal that ComPEFT indeed applies pruning and quantization (**“*ComPEFT applies magnitude pruning and quantizes the remaining weights with a fixed shared value*”**), but **“*its sparsity and quantization strategies differ from ours*.”** Moreover, our primary novelty is not the paradigm of combining pruning and quantization, is the fully data-free pipeline for ultra-efficient delta compression.
> >
> > In summary, ComPEFT and UltraDelta differ in terms of **primary research focus**, **implementation details**, and **data requirements**.
> >
> > ---
> >
> > # Question 2:
> >
> > The reviewer seems to have misunderstood our method. 1) Our **weight variance is used as the metric in the Variance-based Mixed Sparsity Allocation (MSA)**, and **we use entropy to theoretically analyze the effectiveness** of weight variance as a metric. 2) The **activation error** mentioned by the reviewer (Eq.11, line 185) is used in our paper to **explain why high-sparsity models require smaller rescaling factors $\\gamma$**. We derive that the variance of the activation error is related to both sparsity and the additional rescaling factor, and we further empirically show that **the trace norm (sum of singular values, computed without any data)** correlates with the optimal additional rescaling factor $\\gamma$. **Our method does not use data at any stage.**
> >
> > 1. In DAREx, the selection of the rescaling factor is based on the variance refers to the variance of the output activations $c=\\Delta W x$, where $x$ is the input activation and requires data to compute.
> >
> > 2. In our **Variance-based Mixed Sparsity Allocation (MSA)**, we use **weight variance to guide sparsity allocation**, which is computed solely from weights and **does not require data**.
> >
> > 3. The **entropy (Eq.10)** and **activation error (Eq.11)** mentioned by the reviewer are used in the theoretical analyses of two different components: **entropy is used for MSA**, while **activation error is used for Trace-norm-guided Rescaling (TNGR)**. Our Eq.11 indeed follows DAREx, and we explicitly state this in **line 184** of our paper. We further derive **the variance of activation error (Eq.12)** and conclude that **“*As sparsity becomes higher, the variance of the activation error grows rapidly and causes instability*.”** This conclusion **motivates the design of TNGR**, which is **empirically validated and fully data-free**.
> >
> > If we understand correctly, the reviewer thinks our method is not data-free because Eq.11 introduces activation error. However, we emphasize that **this term only appears in our theoretical analysis to explain why we need a smaller rescaling factor for highly sparsed model**. **Our actual method does not require computing activation error, and it is fully data-free**, since both the MSA (based on weight variance) and the TNGR (based on trace norm) are **computed solely from model weights**, without requiring any input data.
> >
> > ---
> >
> > # Question 4:
> > 1. Our determination of the rescaling factor in TNGR **does not depend on input activations or their distribution**. **Eq.11 is only used for theoretical analysis** to show how activation error variance depends on sparsity and the rescaling factor.
> >
> > 2. The actual rescaling factor is **empirically determined** from the trace norm of weights, which is fully **data-free**.
> >
> > 3. We acknowledge that this data-free determination has limitations, and we have explicitly discussed this in the **Limitation** and **Future Work** sections, calling for future research to **design better data-free rescaling factor adjustment methods**.

---

> > ### Author Response · Authors · 2025-08-06
> >
> > We appreciate the reviewer’s time and feedback. We have provided clarifications regarding the raised concern in our previous response. If possible, we would be grateful to know whether our explanation has addressed the reviewer’s question, or if any additional clarification would be helpful.

---

> ### Comment · Reviewer_8YfX · 2025-08-06
>
> ------ Question 1 -------
> Let's stand at the same page, let's refer to the published version of ComPEFT [1]:
>
> -------  designed for PEFT models-----
>
> 1. At the end of ComPEFT abstract : " Lastly, we provide an analysis of different method components, compare ComPEFT with other PEFT methods, and **test its efficacy for compressing full-finetuning residual**", so ComPEFT clearly demonstrate their contribution on **full-finetuning residual**. In addition, they use the whole section 3.3 to demonstrate the effectiveness on **full-finetuning residual**. As a result, their paper has shown enough contribution on full-finetuning residual.
>
> -------- without the need for additional training-------
>
> 2. In their Fig 6, ComPEFT declare larger models do not require explicit tuning of rescaling, as a result, at least for those larger model, they can serve as a data-free baseline.
>
> --------- clarified in the rebuttal that ComPEFT indeed applies pruning and quantization------
>
> 3. ComPEFT does not apply pruning and quantization only at their rebuttal time, their work clearly demonstrate they focus on combining pruning and quantization in their Figure 1.  If you think your contribution is  not the paradigm of combining pruning and quantization, your contribution 1 and 2 will be significantly weakened, thus leading to lack novelty.
>
> [1] ComPEFT: Compression for Communicating Parameter Efficient Updates via **Sparsification and Quantization**.  Published in Transactions on Machine Learning Research (04/2025)
>
> ------- Question 2---------
>
> I'm not satisfied with the author's rebuttal:
>
> 1. **Error of entropy on layer L**:  your Theoretical Analysis of MSA should be the entropy of output error in layer L, otherwise you can not control the entropy using activation error (B.2 Derivation of the Variance of Activation Error).
>
> 2. **Output change Entropy belong to Output change**: Using entropy as a metric for output change $\Delta W x$ is just the same thing as output change itself, especially under this strong gaussian assumption, for example, in the context of massive activation [2], such a assumption is problematic , thus this paper's theory is not only shallow but also weak.
>
> 3. **Wrongly statement** :The author clearly state their theory analysis is based on weight variance and wish to distinguish with DAREx, as state in their  rebuttal "Regarding DAREx[3], we would like to clarify that the variance discussed in DAREx[3] (Theorem 3.1) is not the same as ours. The variance in DAREx[3] refers to activation variance, which requires data to obtain activations. In contrast, our analysis is based on weight variance, which is fully data-free." Now, they say activation only appears in their theoretical analysis, this misalignment raise a concern on their theory analysis.
>
>
> [2] Massive Activations in Large Language Models

---

> > ### Author Response · Authors · 2025-08-06
> >
> > # Question 1:
> > 1. While ComPEFT does include experiments on FFT residuals (only in Section 5.3, not Section 3.3), its **main research focus** remains on PEFT models, as clearly indicated by its name. Our work, on the other hand, specifically targets FFT deltas, making its **primary scope** different from ComPEFT.
> >
> > 2. ComPEFT can only be considered data-free **in specific cases**. However, its general quantization procedure **still relies on a validation set**. In contrast, **our method is fully data-free for all model sizes**, requiring no data under any setting. We design **a truly and fully data-free pipeline applicable to all models**, while ComPEFT’s data-free capability is limited to certain scenarios.
> >
> > 3. We did not claim that the **combination** of pruning and quantization is our novelty. As stated in our paper (Line 115), this paradigm already appeared in previous work such as **DeltaZip**. Our novelty lies in **a fully data-free pipeline for ultra-efficient delta compression**, with each component (Variance-based Mixed Sparsity Allocation, Distribution-Aware Compression, Trace-Norm-Guided Rescaling) providing **new, data-free designs**. The contribution of our work does not rely on the general idea of combining pruning and quantization but on **how we design these components to push the compression–performance trade-offs**.
> >
> > ---
> >
> > # Question 2:
> > We believe there is still a misunderstanding of our theoretical analyses. Our paper contains **two separate theoretical analyses for two different components**:
> > - **Entropy analysis for MSA**: This analysis measures **information content of each layer via weight variance and entropy**. It does not involve activation error or output change; entropy is only used as an information metric to guide sparsity allocation.
> >
> > - **Activation error variance analysis for rescaling factor**: The analysis derives the variance of activation error to explain **why highly sparse models require a smaller rescaling factor**.
> >
> > 1. Our entropy analysis does not involve activation error.
> >
> > 2. Our entropy analysis does not involve output change. Entropy is only used as an information metric to guide sparsity allocation.
> >
> > 3. As we explicitly stated in the rebuttal: “*DAREx studies activation variance for the purpose of rescaling factor adjustment, **our use of weight variance is for layer-wise sparsity allocation***.” Activation only appears in the theoretical motivation for TNGR to explain why highly sparse models require a smaller rescaling factor.

---

> ### Comment · Reviewer_8YfX · 2025-08-08
>
> -----  Question 1 -------
> Still, I'm not satisfied with the author's opinion that :
>
> 1. ComPEFT cannot be served as a important related work just because its "focus on PEFT" due to significant amount of discussion and results on full-finetuning residual, such as end of the abstract, whole section 3.3, section 3.6, Table 4.
>
> 2. As also agreed by authors taht ComPEFT can be considered data-free in certain cases (large LLMs), due to the importance usage of  large LLMs,  should be a important baseline to compare.
>
> 3. The core ideas—layer-wise importance for adaptive pruning [1], delta parameter quantization [2], and rescaling factor adjustments [3]—are each incremental extensions of established compression and pruning techniques. Furthermore, their combination has already been studied in works such as ComPEFT [4].
>
>
> -----  Question 2 -------
>
> I don't think I misunderstand the theory, as state in my original response and post-response, here I summarize my opinion again:
>
> 1. **Theory is not Data-Free** : The authors assert that their analysis is "fully data-free" because it is based on weight variance. However, this is incorrect: their analysis still uses a=Δθx, which inherently depends on the input x, and thus is not data-free. This undermines one of their key claims during rebuttal distinguishing their theory analysis from theory in DAREx.
>
> 2. **Theory cannot support empirical analysis**: This misalignment between empirical and theory also weaken their novelty, and raise potential limitation of its data-free method when x cannot be ignored (e.g. x is large).
>
> 3. **Closely resemble DAREx but shallow**: Although the paper frames its theoretical contributions around bounding entropy, the actual derivation in Appendix B.2 focuses on the activation error 𝜖, and not entropy. This aligns with the approach in DAREx Theorem 3.1, which bounds performance based on output deviation. In fact, the entropy term can be entirely removed from the analysis without affecting the final bound. In addition the paper equates output change with entropy under a strong Gaussian assumption, However, the paper they cite is for vision transformer, common in large language models, this assumption might be violated (e.g., [1]).
>
> 4. **Contradiction in Positioning vs. DAREx** The authors initially claim their method is distinct from DAREx because it is based on weight variance rather than activation statistics. But later, they acknowledge the role of activations in their theoretical analysis. This contradiction introduces confusion and casts doubt on the internal consistency of their theoretical claims.
>
> As a result, due to the misalign between theory and empirical method, theory close resemble to DAREx (as also concerned by Reviewer asSv), the unimportant of entropy analysis, the contribution of theory is limited.
>
> [1]  Massive Activations in Large Language Models
>
>
> Thanks for the author's rebuttal, I decide to maintain my current score and will determine the final rating based on subsequent discussions with other reviewers and the AC.

---

> > ### Author Response · Authors · 2025-08-08
> >
> > We appreciate the reviewer’s comment.
> >
> > ---
> >
> > # Question 1:
> >
> > 1. We have already explained the differences between our work and ComPEFT in our previous response, which is **not limited to the primary research focus**:
> >
> >     >  ComPEFT and UltraDelta differ in terms of **primary research focus**, **implementation details**, and **data requirements**.
> >
> >     In our rebuttal, we also stated that **we will include ComPEFT as a related work** in the final version. Our previous response was meant to answer **“how ComPEFT differs from our method,”** not to suggest that it should not be considered a related work.
> >
> > 2. We have explained in our rebuttal that layer-wise pruning and delta parameter quantization are **broad topics**, and many prior works have explored them. In fact, **delta parameter quantization is discussed in our Related Work section**, and **BitDelta is also one of our baselines**.
> >
> >     What distinguishes our method is not only **the specific design and implementation of each component**, but also the development of **a comprehensive data-free pipeline for ultra-efficient delta compression**:
> >
> >     - Our pruning strategy (**MSA**) is guided by **weight variance** and theoretically analyzed via **entropy**.
> >
> >     - Our quantization-aware pruning (**DAC**) preserves the **intra-layer distributional structure** of delta weights.
> >
> >     - Our rescaling method (**TNGR**) is based on **trace norm** and requires **no data**.
> >
> >      Furthermore, **ComPEFT does not include rescaling factor adjustment**, which is a key part of our method.
> >
> > ---
> >
> > # Question 2:
> >
> > 1. We believe the reviewer **misunderstood the definition of data-free**.
> > - **Data-free refers to pipelines or methods that do not require actual data input to run**, meaning no access to training or validation samples is needed during execution.
> > - It does not **prohibit theoretical derivations from including input variables** like $x$. Such mathematical analysis is conducted at **an abstract level** and does not contradict the data-free nature of our method.
> > - Furthermore, our analysis involving the activation error $\epsilon$ is solely intended as **a theoretical motivation for TNGR**. It is not used in practice, **nor does it require actual data**. The purpose is to **explain why highly sparse models need smaller rescaling factors**.
> >
> > 2. We have validated our method across **a wide range of models, sizes, and tasks**, including scenarios where $x$ is large. Specifically, we evaluate vision models of various scales, NLP models, LLMs of different sizes, and multi-modal models. These evaluations span diverse tasks, consistently demonstrating **the robustness and effectiveness of our method**.
> >
> > 3. We emphasize again that **Appendix B.2 corresponds to TNGR**, not MSA, and therefore it is not directly related to the entropy. **The entropy analysis is part of the theoretical motivation for MSA**. These two analyses are for different components and are clearly separated in the paper.
> >
> >     Regarding the Gaussian assumption, **Figure 6 in DeltaDQ[1]** empirically shows that delta parameters follow Gaussian distributions, which supports our use of this assumption as a reasonable approximation.
> >
> > 4.  Our method **differs from DAREx in several key aspects**:
> >
> > - **Our approach is data-free**, while DAREx requires data for rescaling.
> >
> > - **DAREx uses only random pruning**, whereas our method incorporates both **MSA** (variance-based sparsity allocation) and **DAC** (distribution-aware compression).
> >
> >     In our rebuttal, we clarified that the use of **weight variance** in our method is part of **MSA**, not TNGR:
> >
> >   > “DAREx studies activation variance for the purpose of rescaling factor adjustment, our use of weight variance is for layer-wise sparsity allocation.”
> >
> >     Thus, we did not claim that the key distinction lies in the use of weight variance, but rather in **the overall design and data-free nature of our method**.
> >
> > ---
> >
> > [1] DeltaDQ: Ultra-High Delta Compression for Fine-Tuned LLMs via Group-wise Dropout and Separate Quantization

---

> > ### Author Response · Authors · 2025-08-09
> >
> > Following the reviewer’s suggestion, we have added **ComPEFT baselines on large LLMs (13B)**, following **the experimental settings in Sec. 4.1 (UltraDelta)**.
> >
> > ---
> >
> > For a fair comparison of compression ratios, we adopt **ComPEFT’s storage calculation from Sec. 2.2 (ComPEFT)**, which uses Golomb coding:
> >
> > $\\text{Total Bits} = -\\big[(1-k)\\log_2(1-k) + k\\log_2\\frac{k}{2}\\big] \\cdot d + 16$
> >
> > where $d$ is the total number of parameters.
> >
> > When $d$ is large, the constant term $16$ is negligible, giving:
> >
> > $\\text{Total Bits} \\approx -\\big[(1-k)\\log_2(1-k) + k\\log_2\\frac{k}{2}\\big] \\cdot d$
> >
> > For **UltraDelta**, since we use **4-bit quantization and 97% sparsity** ($k = 0.03$), our total bit calculation becomes:
> >
> > $\\text{Total Bits} \\approx -\\big[(1-k)\\log_2(1-k) + k\\log_2\\frac{k}{16}\\big] \\cdot d$
> >
> > again ignoring the constant term. The **uncompressed model** requires $16 * d$ bits.
> >
> > ---
> >
> > We set **$k = 0.05$, $\\alpha = 1.0$ as ComPEFT’s hyperparameters**, and obtain the following results:
> >
> > |            | Compress | wizardmath-13b | wizardcoder-13b | llama-2-13b-chat | AVG   |
> > | ---------- | -------- | -------------- | --------------- | ---------------- | ----- |
> > |            | 13B      | GSM8K          | HumanEval+      | TruthfulQA       |   -    |
> > | Individual | 1×       | 47.31          | 58.5            | 47       | 50.94 |
> > | ComPEFT    | 47.5×    |    12.21    | 45.5   |   41.23   | 32.98 |
> > | UltraDelta | 50.9×    | **48.45**      | **59.1**        | **48.59**        | **52.05** |
> >
> > UltraDelta achieves **a higher compression–performance trade-off than ComPEFT** under the same evaluation settings. This may be because ComPEFT uses **magnitude pruning**, which as shown in the experiments in our paper, tends to **cause significant performance degradation at high sparsity levels**.

---

### Official Review · Reviewer_asSv · 2025-06-30

**Clarity:** 3
**Significance:** 2
**Originality:** 2
**Rating:** 2
**Confidence:** 4

**Summary:**

This paper proposes UltraDelta, an efficient method to compress delta parameters without additional data. Extensive experiments demonstrate that by assigning layer-wise sparsity rates based on delta parameter variance and rescaling the pruning factor $\gamma$, UltraDelta achieves comparable or better performance to the original fine-tuned model under extreme compression ratios.

**Questions:**

1. DAREx reports that DARE suffers a significant performance degradation when the delta parameters exhibit large mean magnitudes. Does UltraDelta encounter a similar issue?
2. I argue that parameter grouping in the DAC component is unnecessary, as random pruning does not alter the original parameter distribution.
3. Figure 4 employs cosine similarity to assess how well compression methods preserve internal representations. However, despite MP(80×) exhibiting lower final-layer embedding similarity than BitDelta, BitDelta underperforms on the CoLA task (43.51 vs. 36.83, Table 8). A similar pattern is observed for DARE(80×) vs. Delta-CoMe (57.90 vs. 57.67). How should this discrepancy be interpreted?
4. P6 Equation (11), the symbol $\odot$ means Hadamard product or matrix multiplication?

**Ethical Concerns:**

["NO or VERY MINOR ethics concerns only"]

**Final Justification:**

I have read through all reviewers and rebuttals. I choose to maintain my score

- I fully agree with Review 8YfX. The theoretical foundation of the paper lacks rigor, and its primary methodology relies heavily on existing literature, resulting in a lack of originality.

- In the experiments, the content on compression ratios is misleading. I think this is serious issue because it will affect the subsequent work.

Due to the above reasons, I  I think this paper should be rejected.

**Limitations:**

Yes

**Quality:**

2

**Strengths And Weaknesses:**

Strength:

1. The various experiments substantiate that UltraDelta can achieve better or comparable performance with the original fine-tune model even under an extreme compression ratio.
2. The UltraDelta method is efficient because it does not use any additional data.

Weakness:

1. The paper employs sparsification and quantization in its methodology; however, it calculates the compression ratio using techniques from the quantization literature, overlooking the storage implications associated with the positions of non-zero elements. In contrast, conventional sparsification studies typically emphasize the sparsity ratio. Consequently, the actual compression ratio is significantly lower than what is reported.
2. Missing Baseline: The paper does not include a comparison with a simple baseline such as DARE combined with quantization.
3. The method of UltraDelta is not novel. Here is the reason:
  * The idea of  MSA method assigns different sparsity rates to different layers based on their importance score, has been applied in many previous works[1,2].
  * The methodology and analysis of TNGR closely resemble those of prior work [3], which also derived an upper bound for the error $\varepsilon$ and proposed an estimation method for $\gamma$ using auxiliary data. The key distinction between TNGR and DAREx[3] lies in the fact that TNGR does not rely on any additional data. However, the estimation function of $\gamma$ and the setting of the value between 0.5 and 1.0 of TNGR are empirical, which constrains the novelty of TNGR.
4. The models used in the experiments are kind out of dated. More recent models like Qwen2.5 should be taken into consideration.
5. This paper presents experiments on models with easily compressible delta parameters. It is recommended that UltraDelta be evaluated in more challenging scenarios, such as delta parameters with large norms, as indicated in [3].


---
[1]: AlphaPruning: Using Heavy-Tailed Self Regularization Theory for Improved Layer-wise Pruning of Large Language Models

[2]: Adaptive Layer Sparsity for Large Language Models via Activation Correlation Assessment

[3]: Revisiting Delta-Parameter Pruning For Fine-Tuned Models

---

> ### Author Rebuttal · Authors · 2025-07-30
>
> We thank the reviewer for the positive feedback, including: **(i) extensive experiments with strong performance; (ii) an efficient and data-free method**. We now address the reviewer’s questions in detail below.
>
> ---
>
> # Question 1: Large mean magnitude setting
>
> We revisited the DAREx[1] experiments and found that the reported degradation occurs on **MetaMath-Llemma-7B** [8], where DAREx assumed **Llama-2-7B** as the base model. However, according to its huggingFace model card, the correct base model is **Llemma-7B**. We replicated the experiments under both settings:
>
> | Base Model | Finetuned Acc | BitDelta (1bit) | DeltaCoMe (1bit) | DARE (95%) | MP (95%) | UltraDelta (4bit+95%) |
> | - | - | - | - | - | - | - |
> | Llemma-7B  | 63 | 59.89  | 58.18  | 61.93 | 59.65 | **62.40** |
> | Llama-2-7B | 63 | 0.23  | 0 | 0 | 0.91 | 0.91 |
>
> DAREx’s conclusion that **importance-based pruning (such as MP) performs better than DARE[2] in a large mean magnitude setting**, only holds at **low sparsity levels**. At high sparsity, all delta compression methods perform poorly. When using **the correct base model (Llemma-7B)**, all methods perform as expected without severe degradation.
>
> ---
>
> # Question 2: The necessity of DAC
>
> While random pruning may partially preserve the weight distribution, **it does not guarantee robustness**, especially under high sparsity.
>
> 1. **Risk of distortion:** At high sparsity, random pruning can **accidentally distort the original parameter proportions**, making the compressed model **less robust** compared to methods that explicitly maintain weight distribution.
>
> 2. **Empirical evidence:** Our ablation study (Figure 5, Table 9) compares DAC with random pruning across 30 ViT setting. DAC consistently achieves higher accuracy:
> | Method               | 4-bit    | 8-bit     | No quantization |
> | -------------------- | -------- | --------- | --------------- |
> | **DARE**             | 81.58    | 83.85     | 85.88           |
> | **UltraDelta (DAC)** | **86.4** | **90.18** | **92.45**       |
>
>
> These results confirm that **preserving weight distribution explicitly is more robust than random pruning** in large-scale, high-sparsity compression scenarios.
>
> ---
>
> # Question 3: Fig 4
>
> 1. The cosine similarity metric in Figure 4 is intended to illustrate **how well each method preserves the information of the original fine-tuned model, not to predict downstream performance directly**. A higher similarity indicates closer alignment to the uncompressed model’s representations, but **final task accuracy also depends on the uncompressed model’s own performance** on that task.
>
> 2. For example, beyond the two patterns highlighted by the reviewer, we also observe that **UltraDelta outperforms the fine-tuned model (62.64 vs 60.18)**, further showing that **cosine similarity alone does not perfectly reflect downstream performance**.
>
> 3. In the final version, we will include **more figures on other datasets** to better illustrate this relationship and reduce the potential influence of the fine-tuned model.
>
> ---
>
> # Question 4: Eq(11)
>
> It means Hadamard product.
>
> ---
>
> # Weakness 1: Compression ratios
>
> We sincerely thank the reviewer for pointing out this issue.
> 1. Our focus is on exploring **the upper bound of compression ratio achievable with minimal accuracy loss**. We believe this is a meaningful research direction as an ideal compression ratio can **benefit deployment efficiency, reduce communication costs, and storage efficiency**, as Reviewer 9VE5 noted: “*this paper lays a good foundation for system/ML co-design*”.
>
> 2. We acknowledge that a more rigorous evaluation should report the effective compression ratio. Using a **CSR format**, we recalculated the practical compression ratios and found they **remain high while preserving excellent accuracy**:
> | Metric                      | Llama-2-13B | ViT-Base | ViT-Large | T5-Base |
> | --------------------------- | ----------- | -------- | --------- | ------- |
> | Original Accuracy           | 50.94       | **91.0**     | **94.4**      | 86.37   |
> | Compressed Accuracy         | **52.05**       | 90.7     | **94.4**      | **86.74**   |
> | Effective Compression (CSR) | 30.97$\\times$       | 36.92$\\times$    | 109.10$\\times$    | 225.50$\\times$  |
>
> 3. We note that many ongoing works **address the efficiency gap between ideal and practical sparsity representations**, e.g., Block Sparse Row[5], SMaT (SC’24) [6], CS256 (ICLR’25 Workshop on SLLM) [7]. These advances can further improve practical deployment efficiency.
>
> 4. In the revised version, we promise to **clarify our terminology and include both ideal and practical compression ratios** for completeness. We thank the reviewer again for this valuable suggestion that improves the rigor of our work.
>
> ---
>
> # Weakness 2: Missing baseline
>
> We thank the reviewer for this suggestion. **DARE itself does not natively include quantization**, which is why we did not report this baseline in the main paper. Following the reviewer’s comment, we additionally evaluated **DARE + uniform quantization**:
>
> |  | wizardmath-7b | wizardmath-13b | wizardcoder-7b | wizardcoder-13b | llama-2-7b-chat | llama-2-13b-chat |
> | - | - | - | - | - | - | - |
> | Individual  | 41.55 | 47.31 | 49.40 | 58.50  | 45.17 | 47.00 |
> | DARE (4bit+95%)  | 33.78 | 41.23 | 44.20 | 48.50  | 32.12  | 34.40 |
> | UltraDelta (4bit+95%) | **38.76**  | **48.45** | **51.80**  | **59.10** | **46.14** | **48.59**  |
>
> ---
>
> # Weakness 3: Limited novelty
>
> We appreciate the reviewer’s comment. Our novelty lies in **the design, motivation, and strict data-free constraint**.
>
> **1. The first data-free pipeline for ultra-efficient delta compression**
>
> To the best of our knowledge, we are **the first to propose a data-free pipeline for ultra-efficient delta compression**. This makes our approach **practical in scenarios where tuning or data is unavailable**. Our method maintains strong performance under strict data-free constraints, achieving high compression ratios without additional data or tuning.
>
> **2. Novel motivation and design**
>
> Our method is guided by a clear and novel motivation: **maximizing information preservation**. All components are **specifically designed around this motivation**.
>
> - **Variance-based mix sparsity allocation** uses weight variance as a simple, data-free proxy for **information content**, theoretically supported by the variance–entropy relation and rate–distortion theory.
>
> - **Distribution-aware compression** preserves the relative proportions of quantized values to **better preserve the intra-layer information**.
>
> - **Trace-norm-guided rescaling** is designed to estimate an additional rescaling factor using trace norm to stabilize the pruned model based on trace norm, while remaining **fully data-free**.
>
> **3. Clarifications on related works**
>
> - **AlphaPruning[3]** has high computation overhead as it requires **spectral analysis** on every weight matrix, including **SVD/eigenvalue decomposition** and **power-law fitting** to estimate the heavy-tail exponent. ALS [4] requires **data access** to compute activation correlations. In contrast, **our method adopts a simple, data-free variance metric**, offering both **computational efficiency** and **no data dependency**.
>
> - DAREx[1] relies on data (labeled validation or unlabeled), and its own results show that using unlabeled data generally underperforms labeled validation data. However, using labeled validation data may cause the **risk of performance inflation**, which our approach avoids entirely.
>
> - We acknowledge that TNGR are empirically chosen. However, the design is **well-motivated**: the **trace norm captures the effective rank** of delta weights, serving as a simple, **data-free proxy for information content**. We agree this heuristic is a limitation and have explicitly stated it in our ***Limitations*** and ***Future Work*** section, where we plan to conduct **more in-depth research on estimation strategies**, while maintaining the **data-free property** of our approach.
>
> ---
> # Weakness 4: Moer recent LLMs
> We thank the reviewer for this suggestion. We have added experiments on **more recent LLMs**, including **Qwen2.5-7B-Instruct** and **Llama-3.1-Tulu-3-8B-SFT**. We also evaluate them on more challenging tasks:
>
> | Model | Llama | | Qwen | | |
> | - |  - | - | - | - | - |
> | | Humaneval+ | MBPP+ | MATH | GPQA |
> | Finetuned  | 57.3 | 56.6 | 36.2  | 36.87 |
> | BitDelta (1bit) | 55.5  | 52.9 | 38.1 | 37.37 |
> | Delta-CoMe (1bit)  | 53.8 | 53.1 | 29.45 | 31.43 |
> | DARE (95%)  | 50.6 | 50.0 | 37.76  | 34.85 |
> | **UltraDelta (4bit+95%)** |  **56.1**  | **54.5** | **40.14**  | **38.38** |
>
> ---
>
> # Weakness 5: Large norm model
>
> 1. Our experiments already cover a wide range of models of different types, sizes, and compression difficulties, with average L1 norms ranging from **4.1e-5 to 5.01e-3**.
>
> 2. We also conducted tests on the large-norm model reported in DAREx [1] (**MetaMath-Llemma-7B**), as detailed in **Question 1**. When Llama-2-7B was incorrectly used as the base model, the L1 norm reached **1.7e-2**, and in this extreme case **all delta compression methods failed**.
>
> ---
>
> [1] DARE the Extreme: Revisiting Delta-Parameter Pruning For Fine-Tuned Models
>
> [2] Language Models are Super Mario: Absorbing Abilities from Homologous Models as a Free Lunch
>
> [3] AlphaPruning: Using Heavy-Tailed Self Regularization Theory for Improved Layer-wise Pruning of Large Language Models
>
> [4] Adaptive Layer Sparsity for Large Language Models via Activation Correlation Assessment
>
> [5] Automatic Performance Tuning of Sparse Matrix Kernels
>
> [6] High Performance Unstructured SpMM Computation Using Tensor Cores
>
> [7] COMPRESSED SPARSE TILES FOR MEMORY-EFFICIENT UNSTRUCTURED AND SEMI-STRUCTURED SPARSITY
>
> [8] METAMATH: BOOTSTRAP YOUR OWN MATHEMATICAL QUESTIONS FOR LARGE LANGUAGE MODELS

---

> ### Comment · Reviewer_asSv · 2025-08-05
>
> Thank you for the authors' rebuttal. I would like to maintain my score,
> - The authors claim a very high compression ratio in their paper, but the actual ratio is not as impressive. This kind of reporting is problematic, as it can mislead readers and distort fair comparisons with baselines. While the authors addressed this concern in their rebuttal, I believe they have a responsibility to prevent such issues in the original submission.
> - I still believe the novelty of the paper is somewhat limited, as it primarily integrates existing techniques. This concern was also raised by Reviewer 8YfX

---

> > ### Author Response · Authors · 2025-08-05
> >
> > We appreciate the reviewer’s comment. We want to clarify that:
> >
> > ---
> >
> > # Clarification 1: Compression ratio
> >
> > 1. Our method achieves **the best compression–performance trade-off**, delivering **the highest average accuracy among all baselines** and even surpassing the uncompressed fine-tuned model in some cases. Moreover, even after accounting for index overhead, **our actual compression ratios still remain higher than existing methods**, especially on models such as ViT-Large and T5-Base.
> >
> > 2.  We have also discussed **alternative storage formats** beyond CSR (e.g., Block Sparse Row) in our rebuttal, which can **further reduce storage overhead** and increase the actual compression ratio.
> >
> > 3. The primary goal of our work is to **explore the upper limits of delta compression**, as reflected in our title **“*Breaking the Compression Ceiling*”**. This motivates the use of idealized compression ratios to illustrate the maximum achievable compression.
> >
> > 4. We would also like to emphasize that **our original submission already reports the specific quantization bit-width and sparsity rate** for each model in the *Experiment* section. Using the idealized compression ratio was **never intended to mislead readers**, but rather to **highlight the potential compression upper bound achievable by our method**. In the final version, we will **make this point clearer and also provide actual compression ratios** for completeness and fairness in comparison.
> >
> > ---
> >
> > # Clarification 2: Novelty
> >
> > As we clarified in our rebuttal to Reviewer 8YfX, topics such as layer-wise pruning and delta parameter quantization are very broad research areas, and using these terms alone cannot fully characterize our contributions.
> >
> > **1. Novel implementation of each component**
> >
> > - **Variance-based Mixed Sparsity Allocation (MSA)** introduces **weight variance as a simple and data-free metric** for mixed sparsity allocation, theoretically analyzed via entropy.
> >
> > - **Distribution-Aware Compression (DAC)** proposes a **distribution-preserving pruning strategy** that differs significantly from prior importance-based or naive random pruning methods.
> >
> > - **Trace-Norm-Guided Rescaling (TNGR)** provides a **data-free solution** using trace norm, and we have openly acknowledged in our paper that this component is emperical and **may be improved in future work**.
> >
> > **2. The first data-free pipeline for ultra-efficient delta compression**
> >
> > - Our method achieves both **superior compression–performance trade-off** and **full data independence**, that **prior methods have not been able to realize simultaneously**.
> >
> > Given these points, we believe these contributions demonstrate clear novelty at **both the component level and the pipeline level**, beyond simply combining existing techniques. We would also appreciate it if the reviewer could specify which aspects are perceived as non-novel, so we can clarify further.

---

> > ### Author Response · Authors · 2025-08-07
> >
> > We appreciate the reviewer’s time and would like to kindly follow up to ask whether our responses have fully addressed the reviewer’s concerns.
> >
> > As the reviewer noted in the comment, “*the authors addressed this concern (compression ratio) in their rebuttal,*” we have also explicitly committed to clarifying the compression ratio reporting in the final version, and we hope this resolves the issue fairly.
> >
> > If there are still concerns regarding the novelty of our work, we would greatly appreciate it if the reviewer could specify which aspects are perceived as non-novel, so that we can clarify further.

---

### Official Review · Reviewer_vAk5 · 2025-07-01

**Clarity:** 4
**Significance:** 3
**Originality:** 3
**Rating:** 6
**Confidence:** 4

**Summary:**

The paper considers the task of delta compression, where a base model is separately finetuned for several different tasks, and the resulting models have to be stored with as little storage overhead as possible by saving only (a compressed version) of the difference between the base model’s weights and the finetuned weights. The paper proposes UltraDelta, which combines and improves upon two lines of prior work: quantization and pruning. UltraDelta consists of three main improvements over prior methods: (i) a variance-based sparsity allocation (lower sparsity for high-variance layers), (ii) a distribution-aware quantization (uniform quantization followed by group-wise pruning), and (iii) rescaling the quantized weights using the trace norm of the delta weights as a scaling factor. The results across a wide range of models, tasks, and domains show that UltraDelta achieves very high compression ratios (up to 800x) while maintaining (sometimes even increasing) strong performance.

**Questions:**

* Intuitively, why is it important to “maintain the relative proportions across different quantized weight values”? Is there a good explanation for why this should achieve better performance than, e.g., uniform or magnitude-based pruning?

**Ethical Concerns:**

["NO or VERY MINOR ethics concerns only"]

**Final Justification:**

The authors have sufficiently addressed my concern about the method's efficiency, which is why I am increasing my score.

**Limitations:**

Yes

**Paper Formatting Concerns:**

There are no major formatting issues in this paper.

**Quality:**

4

**Strengths And Weaknesses:**

**Strengths**

* The paper conducts an extensive empirical validation of the proposed method across models of different scales, types, and tasks (LLMs, vision models, and multi-model models), leaving no doubt about the superiority of the proposed approach.
* The paper is very well-written and easy to follow. Figure 2 is a work of art and helps the reader understand the different components of the proposed method.
* The proposed method addresses key weaknesses of prior approaches: (i) treating all layers equally, (ii) disrupting intra-layer weight distributions, and (iii) instability under high compression.
* The paper provides good theoretical motivation and empirical validation, including ablation studies, for the proposed method.

**Weaknesses**

* The paper claims that its method does not affect the inference stage, which is not entirely true. Concretely, at inference time, the delta weights need to be added to the base model’s weights, which introduces a (non-negligible?) memory and compute overhead, especially for large models.

---

> ### Author Rebuttal · Authors · 2025-07-29
>
> We sincerely thank the reviewer for the constructive feedback and positive comments on our work, including: **(i) Extensive experiments; (ii) Clear and well-written; (iii) Addresses prior weaknesses; (iv) Strong theoretical and empirical validation**. We are especially grateful for the kind recognition of **Figure 2**, described as *“a work of art”*. As our method consists of multiple components, we invested great effort in designing this figure to help readers quickly grasp our approach, and we are delighted that this effort was appreciated.
> We now address the reviewer’s comments in detail below.
>
> ---
> # Question 1 : Explanation for maintaining relative proportions across values
>
> We thank the reviewer for the question. Below, we explain why maintaining relative proportions across different quantized values is important, supported by **theoretical, methodological, and empirical perspectives**.
>
> **1. Theoretical perspective**
>
> - Our goal in maintaining the relative proportions of different quantized values is to **keep the weight distribution before and after pruning as similar as possible**. This can be formalized using **KL divergence** between the original quantized distribution $P = \\{p_i\\}$ and the pruned distribution $Q = \\{q_i\\}$:
>    $$
>    D_{\\mathrm{KL}}(P \\parallel Q) = \\sum_{i} p_i \\log \\frac{p_i}{q_i}
>    $$
> where $p_i$ and $q_i$ denote the proportion of the $i$-th quantized value before and after pruning. When the relative proportions of quantized values are better maintained, the pruned distribution $Q$ stays closer to the original quantized distribution $P$, resulting in a lower $D_{\\mathrm{KL}}$. **A lower $D_{\\mathrm{KL}}$ means the weight distribution is better preserved**.
> - Preserving the weight distribution is important because it **reflects the statistical structure of the model parameters**. Weights are not just arbitrary numbers; their distribution encodes **structural and semantic information** learned during training. Preserving distribution helps maintain model behavior after compression.
> - Furthermore, **Figure 4** provides empirical evidence supporting this claim: it shows that **UltraDelta achieves higher cosine similarity** between the compressed model’s outputs and those of the original model, showing that **preserving weight proportions helps retain the semantic information of the original model**.
>
> **2. Method comparison**
> - **DAC (ours)** uses group-wise pruning to explicitly preserve the relative proportions of quantized values, **maintaining the overall weight distribution**. This approach is **highly robust**, avoiding the randomness that could distort proportions under high sparsity.
> - **Magnitude pruning** removes weights with smaller magnitudes, **shifting the distribution toward large values**. This **ignores the balancing effect of small weights**. The imbalance weakens parameter interactions and leading to accuracy degradation.
> -  **Uniform pruning** removes weights randomly and can partially keep the distribution. However, under high sparsity it may still accidentally distort the distribution, showing **lower robustness** compared to methods that explicitly preserve weight proportions.
>
> **3. Experiment validation**
> - As shown in our experiments, under high sparsity, **magnitude pruning performs worse than almost all other methods**, showing that **disrupting the weight distribution can cause severe performance degradation**.
> |  Method      | LLM-7B    | LLM-13B   | T5-Base   | ViT-Large |
> | --------------------- | --------- | --------- | --------- | --------- |
> | **Magnitude Pruning**     | 14.19     | 7.40      | 56.08     | 11.20     |
> | **UltraDelta (Ours)** | **45.57** | **52.05** | **86.74** | **94.40** |
>
> *Note: These results are extracted from our Experiment section. Both magnitude pruning and UltraDelta are under the same compression ratios.*
>
> - Our **ablation study (Figure 5 and Table 9)** compare DAC against uniform/random pruning across 30 ViTs compression settings. In such large-scale compression scenarios, robustness is particularly important. The results show that **DAC is significantly more robust than uniform pruning**, confirming the effectiveness of preserving weight distribution.
> | Method               | 4-bit    | 8-bit     | No quantization |
> | -------------------- | -------- | --------- | --------------- |
> | **DARE**             | 81.58    | 83.85     | 85.88           |
> | **UltraDelta (DAC)** | **86.40** | **90.18** | **92.45**       |
>
> *Note: This table is extracted from the experimental results reported in our paper (Table 9, visualization in Figure 5), comparing DARE[1] and DAC under different quantization settings (4-bit, 8-bit, and no quantization).*
>
> ---
>
> # Weakness 1 : Inference Overhead
>
> 1. Our statement was in comparison with **other delta compression methods**, which also require adding delta weights at inference time. Our method only needs a **simple addition**, while methods like **Delta-CoMe[2] involve matrix multiplications** to project low-rank matrices back into the full parameter space before inference, which are more expensive.
> 2. The addition can be done on **CPU** before inference, so it **does not increase GPU memory usage** and introduces **minimal compute overhead**. For example, we measured total inference time on WizardCoder-Python-7B-V1.0 evaluated on HumanEval+. We compared the **fine-tuned model, UltraDelta (addition on CPU), and Delta-CoMe (layer-wise matrix multiplication on CPU)**:
> | Method  | Fine-tuned | UltraDelta | Delta-CoMe |
> | ------- | ---------- | ---------- | ---------- |
> | **Runtime** | 217s       | 221s       | 341s       |
>
> 3. We will **clarify this in the revised version** and add inference time comparisons. Thank you for helping us improve the precision of our claims.
>
> ---
>
> [1] Language Models are Super Mario: Absorbing Abilities from Homologous Models as a Free Lunch
>
> [2] Delta-CoMe: Training-Free Delta-Compression with Mixed-Precision for Large Language Models

---

> > ### Comment · Reviewer_vAk5 · 2025-08-06
> >
> > Thank you for the response, which fully clarifies my concerns. I will increase my score accordingly.

---

> > > ### Author Response · Authors · 2025-08-07
> > >
> > > Thank you for your comment and support for our work. We also sincerely appreciate your kind words about Figure 2, as well as the valuable suggestions you provided, which helped improve the rigor of our paper.

---

### Official Review · Reviewer_9VE5 · 2025-07-01

**Clarity:** 3
**Significance:** 3
**Originality:** 4
**Rating:** 6
**Confidence:** 4

**Summary:**

This paper presents UltraDelta, the first data-free delta compression pipeline that claims to achieve both ultra-high compression and strong (accuracy) performance. There are three key components in UltraDelta: 1) Variance-Based Mixed Sparsity allocation; 2)  Distribution-Aware Compression; 3) Trace-Norm Guided Rescaling. The resulting compression ratios across multiple different categories of models are impressive: 133x for LLMs; 800x for general NLP models; 400x for ViT models; and 40x for multi-modality models.

**Questions:**

- In the “Theoretical Analysis of MSA”, the authors explicitly focused on sparsity by saying “...should be assigned smaller sparsity”. I wonder if the same conclusion holds for quantization? For instance can we say: “if a layer exhibits larger standard deviation, we should use higher bit width”?
-  I am also wondering if the variance could be a good signal for selecting sparsity level / bitwidth? So instead of saying: “should be assigned smaller sparsity”, can we suggest: “should be assigned a sparsity in the range [a,b]?”

**Ethical Concerns:**

["NO or VERY MINOR ethics concerns only"]

**Final Justification:**

I maintain my score and I think this is overall a good paper.

I don't understand the comment on novelty. If we execute the standard that "if this is done by someone else in some field, then it is not novel", we should reject most of the papers, including the paper that proposed deep learning since it is merely an application of chain rule.

Some reviewers raised the baseline of ComPEFT, and I think the authors have provided evidence that their approach outperforms it, quite significantly, which is an evidence of usefulness of this work to the community.

**Limitations:**

I think it is fair to mention this work is primarily on compression ratio and serving performance improvement could be a future work.

**Quality:**

4

**Strengths And Weaknesses:**

# Strength:
- The problem itself is important to not only the ML research community but also the system & MLSys community. It is also a timely and very relevant topic. As someone working on delta compression in the past, some of the questions that the authors try to resolve are the issues asked by reviewers in our previous papers. Overall I think this paper is a nice addition to the NeurIPS 2025 program.

- The paper is very well written and easy to follow. I particularly like the figures across the paper as they are easy to read and accessible.
- It’s great to see the authors provide not only empirical results, but also theoretically analyze the effect of MSA. I think it also suggests why model deltas are more compressible than the entire models (based on the findings from DeltaZip & DeltaDQ, the variance or standard deviation in model deltas are significantly smaller than the entire model). I have a few suggestions & follow-up questions regarding this (please see questions & weakness).
- The experiments conducted are extensive, covering a wide range of tasks / models including not only LLMs, but also other NLP models &

# Weaknesses

- [NIT]: In “Theoretical Analysis of MSA”, I wonder if the authors could show some empirical results as well. For example, say we want to maintain 99% accuracy/cosine similarity, and x-axis is the standard deviation, y-axis is the sparsity we can achieve.
- [Follow Up on above & NIT]: I think this entropy explanation is nice -- however I think there're two potential improvements: 1) If I understood it correctly, the entropy viewpoint suggests that if you apply lossless compression, then the lower the variance, the lower bit rate you can achieve. However, across your paper you are actually apply lossy compression. I think another viable approach is to try rate-distortion theory. 2) However these theories only give you lower bound -- and it does not necessarily hold for the exact compression method you used here -- some empirical results would enhance it. See [this blog post (Why Delta Compression Works)](https://about.yao.sh/posts/why-delta-works/) as a reference.

- One of the main weaknesses is that it’s unclear how efficient the proposed approach will be, when deployed in a real serving engine, i.e., whether the compression ratio can be translated well into serving performance (throughput; latency) improvements.

- An example with LLMs: BitDelta quantizes the model delta to 1 bit without sparsity vs. UltraDelta quantizes/sparsifies the model delta to 4 bit + 95% of sparsity -- which one is faster? Although UltraDelta achieves higher compression ratio, it won’t be faster without a properly implemented kernel to achieve the highly sparse computation. Serving performance is also a key focus in DeltaZip and therefore they chose to use structured sparsity, which only gives 50% sparsity. Ultimately, we want faster inference, instead of just high compression ratio.
- I don’t think this should be a show-stopper, as this paper does not focus on serving performance, however, I think it is fair to mention this in the limitations section to give enough context to the readers. Ultimately I think this paper lays a good foundation for system/ml co-design.

- I would also strongly suggest authors to place this work into more context in the introduction, as the ultimate goals of these works (DeltaZip, BitDelta, DeltaDQ, etc) are to improve deployment efficiency, UltraDelta offers a significant higher compression ratio which can be leverage to further improve the deployment efficiency.
- [NIT] I am a bit surprised the proposed approach consistently outperforms the uncompressed baseline. Although I acknowledge there might be some implicit regularization effect in compression as described by the authors -- I think some more empirical results on more controlled experiments (e.g., those models are fine-tuned by the authors with varying steps, and then compressed by the authors - to show if the models are overfitting/underfitting or not) can be beneficial.
- Re: Hyper-parameter selection. I wonder how the authors select the sparsity level in their experiments. For example, with LLMs the authors used 95%/97% sparsity; 99% for other NLP models; 90% for multi-modality models. Do the authors perform an extensive (but might be expensive) grid search of these parameters in their experiments? Or are there any other signals used in the process?
- For LLMs, I think some more challenging tasks should be considered in the evaluations, such as AIME/GQA, etc.

---

> ### Author Rebuttal · Authors · 2025-07-30
>
> We sincerely thank the reviewer for strong support of our work, including: **(i) important and timely topic; (ii) well written and easy to follow; (iii) theoretical and empirical analysis; (iv) extensive experiments**. We are especially grateful for **the many constructive suggestions and insightful questions**, which significantly helped us improve the depth of our paper.
>
> ---
>
> # Question 1: Variance-Bitwidth Relationship
>
> We thank the reviewer for the insightful question. We have observed that **variance can also serve as a metric for bit-width allocation**.
> 1. According to the variance–entropy relationship and the rate–distortion theory, **variance is theoretically linked to compression rate**, which depends on both sparsity and bit width.
> 2. We empirically validated this in 8 ViT-B/32, where bit width was allocated based on layer variance (4-bit to layers with larger variance, 2-bit to layers with smaller variance, together it obtains 3-bit/parameter quantization). The **mixed 3-bit setting achieved accuracy close to uniform 4-bit quantization**, showing that variance is an effective metric for bit width allocation.
> |  | SUN397 | Cars | Resisc45 | EuroSAT | SVHN | GTSRB | MNIST | DTD  | AVG  |
> | - | - | - | - | - | - | - | - | - | - |
> | Finetuned | 79.2   | 77.7 | 96.1 | 99.8 | 97.4 | 98.7 | 99.7 | 79.4 | 91.0 |
> | UltraDelta (uniform 4bit) | 78.6 | 77.4 | 95.7 | 99.4 | 97   | 98.7  | 99.7  | 79  | 90.7 |
> | UltraDelta (uniform 3bit) | 77.6 | 75.7 | 95.3 | 99.3 | 96.8 | 98.6  | 99.7  | 78.7 | 90.2 |
> | UltraDelta (mix 3bit)  | 78.2 | 77.2 | 95.6 | 99.3 | 96.8 | 98.8  | 99.6  | 79.0 | 90.6 |
>
> ---
>
> # Question 2 : Can variance serve as a signal for sparsity/bitwidth
>
> While **variance provides a strong signal for allocating *relative* sparsity levels or bit widths**, it is **not sufficient to select precise numerical values** for either parameter. Our method uses **layer-wise variance** to guide allocation, but evaluating per-layer performance is impractical in experiments. Therefore, our **analysis is conducted at the model level**, and the standard deviation is calculated based on the overall weight of the model.
>
> **1. Bit width:**  We conducted experiments to test whether variance can guide the selection of an optimal quantization bit width. Specifically, we quantized 8 ViT-B-32 models using different bit widths (without sparsity, **only quantization**), due to space limits, we only show some of the results.
> - Across all datasets, **2-bit quantization is optimal**, retaining ~99% of the finetuned model performance.
> - Bit width is a **discrete variable with large step size**. Performance drops drastically from 2-bit → 1-bit.
> - Therefore, variance cannot directly predict this optimal discrete value, as **the available discrete bitwidth levels have large gaps and lack fine granularity for precise selection**.
>
>    |  | SUN397 | Cars  | Resisc45  | EuroSAT | SVHN | GTSRB | MNIST | DTD | AVG |
>    | - | - | - | - | - | - | - | - | - | - |
>    | Std  | 0.000312  | 0.000299  | 0.000271  | 0.000242  | 0.000288  | 0.000250  | 0.000261  | 0.000263  | -  |
>    | Finetuned | 79.2 | 77.7 | 96.1 | 99.8  | 97.4 | 98.7 | 99.7 | 79.4 | 91.0  |
>    | 1-bit  | 65.8  | 62.3 | 68.0 | 58.8  | 37.9 | 39.3 | 59.6  | 49.3 | 55.1 |
>    | 2-bit  | 79.2  | 77.6 | 95.5 | 99.1 | 96.8 | 98.3  | 99.6  | 79.3  | 90.7  |
>    | Optimal Bitwidth | 2 | 2 | 2 | 2 | 2 | 2 | 2 | 2 | 2 |
>
> **2. Sparsity level:**  We conducted experiments on sparsity selection using different sparisty (using **only random pruning**). The optimal sparsity is the highest sparsity that preserves ~99% of the original accuracy.
> - The results show that for models with **lower original accuracy (harder tasks)**, **layers with larger variance tend to require lower sparsity**.
> - For models with **higher original accuracy (easier tasks)**, variance has **a weaker impact** on the optimal sparsity level.
> - Therefore, variance is correlated with sparsity and **serves as a reliable signal for allocating relative sparsity levels across layers**. But **the precise sparsity value for the whole model can be influenced by task complexity**. However, **task complexity is consistent across layers of a single model**, making variance a robust metric for relative layer-wise sparsity allocation.
> |  | SUN397 | Cars | Resisc45 | EuroSAT  | SVHN | GTSRB | MNIST    | DTD  |
> | - | - | - | - | - | - | - | - | - |
> | Finetuned | 79.2 | 77.7 | 96.1 | 99.8 | 97.4  | 98.7 | 99.7 | 79.4 |
> | std | 0.000312 | 0.000299 | 0.000271 | 0.000242 | 0.000288 | 0.000250 | 0.000261 | 0.000263 |
> | Optimal Sparsity | 0.88  | 0.89  | 0.97  | 0.995  | 0.99  | 0.995 | 0.995  | 0.92  |
>
> ---
>
> # Weakness 1 : Analysis of MSA
>
> We thank the reviewer for these valuable comments. We address the two points as follows:
>
> **1. Empirical upper bound**
>
> The results can be found in the answer to Question 2. These results show that, **at the model level, the empirical upper bound is not solely determined by variance; it also depends on task difficulty**. We will include this analysis and figure in the revised version.
>
> **2. Theoretical analysis**
>
> We initially adopted an entropy-based discussion because entropy is a well-established proxy for the **information content**. It provides a simple, intuitive link between weight variance and the amount of information in each layer.
>
> We fully agree that **rate–distortion theory is a more appropriate framework** under lossy compression settings like pruning, as it explicitly connects **variance to information rate and distortion**. A simplified version of the proof is provided **in Reviewer 8YfX’s Question 2** (omitted here due to space limits), and we will include a more detailed proof in the revised version of the paper.
>
> Thanks to the reviewer again for helping us strengthen the analysis of MSA.
>
> ---
>
> # Weakness 2 : Deployment Efficiency
>
> 1. We thank the reviewer for highlighting the importance of deployment efficiency, which is indeed a primary objective of many delta compression works such as **DeltaZip**.
> 2. We acknowledge that UltraDelta is **not designed to directly optimize deployment efficiency**, and achieving real inference acceleration typically requires specialized kernels or structured sparsity. Our focus in this paper is orthogonal: we aim to explore **data-free delta compression** methods that maximize information preservation, **providing a foundation that can potentially complement future system-level optimizations**.
> 3. Following the reviewer’s suggestion, we will **explicitly state this limitation** in the revised version and **clarify in the introduction** that works such as DeltaZip primarily target deployment efficiency.
>
> ---
>
> # Weakness 3 : Controlled Experiment
>
> We sincerely thank the reviewer for this insightful comment. To investigate this, we conducted additional controlled experiments on two datasets, **SUN397 and Cars**, using a pretrained ViT-B/32 model. We trained the models **with varying fine-tuning steps** and then applied **UltraDelta**. The results show that:
>
> 1. **Underfitted models** (early training steps, higher loss) consistently benefit from UltraDelta, exhibiting **noticeable accuracy improvements** compared to the baseline.
> 2. **Well-fitted models** (later steps, lower loss) do not show significant performance gains after compression. **In SUN397**, we observe only a small improvement after compression, possibly because **the task is more complex and data-rich**, making it harder for the model to fully fit the dataset.
> 3. The observed effect mainly reflects that **UltraDelta benefits models that are underfitted**, while well-trained models already capture most of the task information, leaving less room for improvement.
>
> | Training Step | 1000  | 2000 | 3000 | 4000 | 5000 | 6000 | 7000 | 8000 | 9000 | 10000 |
> | - | - | - | - | - | - | - | - | - | - | - |
> | **Loss (Cars)**  | 0.1404 | 0.0834 | 0.0206 | 0.0069 | 0.0059 | 0.0059 | 0.0034 | 0.0077 | 0.0048 | 0.0036 |
> | **Acc (Cars)** | 72.86  | 68.90  | 75.22  | **78.78**  | **77.74**  | **78.37**  | **78.01**  | **75.21**  | **77.60**  | **77.34**  |
> | **UltraDelta (Cars)** | **74.02**  | **73.27**  | **75.82**  | 75.56  | 75.62  | 75.66  | 76.13  | 72.80  | 74.74  | 74.06  |
> | **Loss (SUN397)**  | 0.5475 | 0.1432 | 0.0426 | 0.1125 | 0.0526 | 0.0040 | 0.0018 | 0.0009 | 0.0006 | 0.0004 |
> | **Acc (SUN397)**  | 73.63 | 75.94 | 75.40 | 74.63 | 74.07 | **76.94** | 76.87 | 77.03 | 77.06 | 77.07 |
> | **UltraDelta (SUN397)** | **76.02** | **76.92** | **77.21** | **76.20** | **76.19** | 76.66 | **76.94** | **77.11** | **77.19** | **77.31** |
>
> This explains why performance improvements are more commonly observed on **LLMs**, as they are typically evaluated in a **zero-shot setting** without task-specific fine-tuning, making them more likely to benefit from the regularization effect of compression.
>
> ---
>
> # Weakness 4 :  Hyper-parameter Selection
>
> Our choice of sparsity levels is guided by prior findings in delta compression, which show that models can typically have **around 90% sparsity** without severe accuracy degradation. Based on this, we explored a **coarse set of candidate values** (90%, 95%, 99%, 99.9%) to establish a reasonable range, and then made **minor adjustments** for specific models when accuracy changed a lot within this range.
>
> ---
>
> # Weakness 5 : More Challenging Tasks
>
> We expanded the LLM evaluations to include more challenging tasks: **MATH** (with some AIME problems), **GPQA**, **MBPP+**, and **HumanEval+**, using updated models **Qwen2.5-7B-Instruct** and **Llama-3.1-Tulu-3-8B-SFT**. Results are shown below:
>
> | Model  |  Llama  |  | Qwen |  |
> | - | - | - | - | - |
> |  | Humaneval+   | MBPP+  | MATH  | GPQA |
> | Finetuned | 57.3  | 56.6 | 36.2  | 36.87  |
> | BitDelta (1bit)  | 55.5  | 52.9  | 38.1  | 37.37  |
> | Delta-CoMe (1bit)  | 53.8  | 53.1  | 29.45  | 31.43  |
> | DARE (95%)  |  50.6 | 50.0  | 37.76 | 34.85  |
> | UltraDelta (4bit+95%) |  **56.1** | **54.5** | **40.14**  | **38.38** |

---

### Comment · Area_Chair_toLW · 2025-08-06

Dear Reviewers,

This is a quick reminder that we are now in the post-rebuttal discussion phase.

Please take the time to read the author rebuttal and engage in discussion with the other reviewers. Your input is crucial for us to make a final, informed decision as the deadline is approaching.

Thank you for your timely participation.

Best regards,

Area Chair

---

### Note · Authors · 2025-08-13

We sincerely thank all reviewers, ACs, SACs, and PCs for their time, constructive feedback, and support. We now summarize our strengths and key rebuttal highlights as follows:

---

**Key Strengths Highlighted by Reviewers**

1. **Efficient** and **data-free**. (asSv)
2. **Addresses key weaknesses** of prior approaches: (i) treating all layers equally, (ii) disrupting intra-layer weight distributions, and (iii) instability under high compression. (vAk5)
3. **Extensive experiments** across models of different scales, types, and tasks, showing superior performance. (9VE5, vAk5, asSv, 8YfX)
4. **Well-written** and easy to follow, with clear and informative figures. (9VE5, vAk5, 8YfX)
5. Provides **both empirical validation and theoretical analysis**. (9VE5, vAk5)
6. **Tackles an important problem (delta compression)** relevant to both the ML research community and the MLsys community. (9VE5, 8YfX)

---

**Key Rebuttal Highlights**

1. **Novelty:** We clarified that UltraDelta is the **first data-free pipeline for ultra-efficient delta compression**, composed of three **newly designed components** at inter-layer, intra-layer, and global levels. (asSv, 8YfX)
2. **Compression Ratio:** We reported **actual CSR-based compression ratios**, showing that UltraDelta achieves **the best compression–performance trade-off**. (asSv)
3. **Theoretical Analysis:** We enriched the theoretical analysis by adding **rate–distortion theory** for MSA and providing a **systematic explanation** of DAC's effectiveness. (9VE5, vAk5, 8YfX)
4. **Harder Tasks & Newer LLMs:** We added experiments on **Qwen2.5 and Llama-3.1**, evaluated on challenging tasks **MATH and GPQA**. (9VE5, asSv)
5. **Additional Baselines:** We included **DARE + uniform quantization** as an additional baseline, and added **ComPEFT** comparisons to large LLMs (13B). (asSv, 8YfX)
6. **Inference Overhead:** We showed that UltraDelta’s dequantization and addition operations can be **efficiently executed on CPU with negligible latency**. (vAk5)

---

We believe **UltraDelta makes critical contributions**: (i) it is the **first data-free pipeline for ultra-efficient delta compression**, (ii) introducing **three newly designed components**, and (iii) being supported by **extensive experiments** and **strong theoretical motivation**. We also commit to add all clarifications and suggestions raised during the rebuttal into the final version.

We thank all reviewers, ACs, SACs, and PCs again for their valuable efforts.

---

### Decision · Program_Chairs · 2025-09-17

**Decision:**

Accept (poster)

**Comment:**

(a) Summary of Scientific Claims and Findings

This paper introduces UltraDelta, a novel data-free pipeline for the delta compression of fine-tuned models. The primary goal is to address the significant storage overhead associated with storing numerous task-specific models. The proposed method consists of three key components designed to work in concert: (1) Variance-Based Mixed Sparsity Allocation (MSA), which assigns layer-wise sparsity based on weight variance to preserve information in high-variance layers; (2) Distribution-Aware Compression (DAC), which applies uniform quantization followed by group-wise pruning to maintain the intra-layer weight distribution; and (3) Trace-Norm-Guided Rescaling (TNGR), which uses the trace norm of delta weights to estimate a global rescaling factor for model stability. The authors claim that this pipeline achieves state-of-the-art compression ratios while maintaining strong performance across a wide variety of models, including LLMs, vision transformers, and general NLP models, all without requiring any data for the compression process.

(b) Strengths of the Paper

The paper has several notable strengths, as highlighted by the reviewers:

Practical Importance: It addresses the highly relevant and practical problem of model storage, which is a significant challenge in the era of large-scale model fine-tuning.

Data-Free Approach: The core contribution of a completely data-free compression pipeline is a major practical advantage, making the method highly applicable in real-world scenarios where data may be unavailable due to privacy, cost, or accessibility constraints.

Extensive and Strong Empirical Validation: The authors provide comprehensive experiments across a diverse set of models (LLaMA-2, RoBERTa, ViT), scales, and tasks. The results consistently demonstrate that UltraDelta outperforms existing methods, especially at very high compression ratios.

Clarity and Presentation: The paper is generally well-written, clearly structured, and easy to follow. The figures, particularly the one illustrating the pipeline, are effective in conveying the method's components.

(c) Weaknesses of the Paper

Despite its strengths, the paper has some weaknesses that were pointed out during the review process:

Limited Novelty: A primary concern, raised by multiple reviewers (asSv, 8YfX), is the novelty of the individual components. The core ideas of using layer-wise importance for pruning, quantization, and rescaling factors are not new. While their combination into a data-free pipeline is the main contribution, some reviewers felt the work was an incremental extension of established techniques.

Initial Misleading Metrics: The original submission reported idealized compression ratios that did not account for the storage overhead of sparse matrix indices. This is a significant omission that inflates the perceived performance. While the authors corrected this in the rebuttal, it was a flaw in the initial manuscript.

Theoretical Foundation: The theoretical justifications were found to be somewhat weak and not rigorously proven by one reviewer (8YfX). The analysis could have been better situated within and built upon existing theoretical frameworks from prior work.

(d) Reasons for Decision

My recommendation is to accept this paper as a poster. The decision is based on the significant practical value and strong empirical results of the proposed UltraDelta pipeline. While the concerns about the incremental nature of the technical novelty are valid, the paper's main contribution lies in the successful design and integration of these components into a fully data-free framework that demonstrably pushes the state-of-the-art in delta compression. The extensive experiments provide compelling evidence of the method's effectiveness and broad applicability. The data-free aspect, in particular, is a strong selling point that makes this work a valuable contribution to the community. A poster presentation would be an excellent venue for the authors to share their practical results and engage in discussions about the trade-offs of their approach.

(e) Summary of Rebuttal Period

The rebuttal period featured a lively discussion. Reviewers raised several key points, including concerns about novelty, the calculation of compression ratios, the theoretical underpinnings, and missing baselines.

The authors were responsive and provided a detailed rebuttal. They conducted new experiments on more recent models (Qwen2.5, Llama-3.1) and more challenging tasks, added the requested baseline (DARE + uniform quantization), and, importantly, recalculated the compression ratios using a more realistic CSR format, which addressed a major concern.

However, the discussion surrounding novelty became contentious. The authors' defense of their work, particularly against claims that it was too similar to prior art like ComPEFT, was somewhat rigid. For instance, when reviewers pointed out similarities, the authors' rebuttal focused on differentiating their work rather than acknowledging the conceptual overlap and then providing a detailed, nuanced comparison to highlight their specific innovations. A more effective approach would have been to graciously accept the connection to the related work and then clearly articulate what makes their variance-based metric superior to magnitude-based pruning in a data-free context, or how their trace-norm rescaling is a novel data-free alternative. This would have been more constructive and persuasive.

In my final decision, I have weighed the authors' thoroughness in addressing the empirical concerns more heavily than the lingering debates on novelty. The paper is empirically sound and presents a useful method. However, I strongly urge the authors to revise the final version to incorporate a more balanced and detailed discussion of the related work mentioned by the reviewers. Acknowledging the context and clearly carving out their specific contribution will significantly strengthen the paper.